# Ultrastrong magnon-magnon coupling and chiral spin-texture control in a dipolar 3D multilayered artificial spin-vortex ice

Troy Dion [1] ✉, Kilian D. Stenning [2,3,4], Alex Vanstone [2], Holly H. Holder [2], Rawnak Sultana [5], Ghanem Alatteili [6], Victoria Martinez [6], Mojtaba Taghipour Kaffash [5], Takashi Kimura [1], Rupert F. Oulton [2], Will R. Branford [2,4], Hidekazu Kurebayashi [3,7,8], Ezio Iacocca [6], M. Benjamin Jungfleisch [5] & Jack C. Gartside [2,4] ✉

Strongly-interacting nanomagnetic arrays are ideal systems for exploring reconfigurable magnonics. They provide huge microstate spaces and integrated solutions for storage and neuromorphic computing alongside GHz functionality. These systems may be broadly assessed by their range of reliably accessible states and the strength of magnon coupling phenomena and non-linearities. Increasingly, nanomagnetic systems are expanding into three-dimensional architectures. This has enhanced the range of available magnetic microstates and functional behaviours, but engineering control over 3D states and dynamics remains challenging. Here, we introduce a 3D magnonic metamaterial composed from multilayered artificial spin ice nanoarrays. Comprising two magnetic layers separated by a non-magnetic spacer, each nanoisland may assume four macrospin or vortex states per magnetic layer. This creates a system with a rich $16^N$ microstate space and intense static and dynamic dipolar magnetic coupling. The system exhibits a broad range of emergent phenomena driven by the strong inter-layer dipolar interaction, including ultrastrong magnon-magnon coupling with normalised coupling rates of $\frac{\Delta f}{\nu} = 0.57$, GHz mode shifts in zero applied field and chirality-control of magnetic vortex microstates with corresponding magnonic spectra.

Artificial spin systems and related nanomagnetic arrays provide a rich environment for engineering functional magnetic behaviours[1–6]. Comprising many strongly interacting magnetic nanoislands, these metamaterial systems offer great freedom in designing magnetic behaviours via array geometry and nanoisland shape. The many-body interactions give rise to complex emergent behaviours and vast microstate spaces ($2^N$ in an $N$-element system for traditional macrospin islands). One fertile area of exploration in such systems is magnonics[7]. Magnons are the quanta of magnetic spin-waves, collective oscillations of the magnetic texture. The manipulation of magnons to transmit and process information is termed magnonics[3,7,8]. The limiting factors on a system's magnonic efficacy are the range of readily accessible states and the strength of magnon interactions within the system. Artificial spin systems offer an

[1]Solid State Physics Laboratory, Kyushu University, Fukuoka, Japan. [2]Blackett Laboratory, Imperial College London, London, UK. [3]London Centre for Nanotechnology, University College London, London, UK. [4]London Centre for Nanotechnology, Imperial College London, London, UK. [5]Department of Physics and Astronomy, University of Delaware, Newark, DE 19716, USA. [6]Center for Magnetism and Magnetic Nanostructures, University of Colorado Colorado Springs, Colorado Springs, CO 80918, USA. [7]Department of Electrical and Electronic Engineering, University College London, London, UK. [8]WPI Advanced Institute for Materials Research, Tohoku University, Sendai, Japan. ✉e-mail: troy.dion@phys.kyushu-u.ac.jp; j.carter-gartside13@imperial.ac.uk

extremely rich state space, but typically struggle to provide higher intensity magnon coupling.

Generally, strong magnon interactions are obtained at the expense of system reconfigurability. Examples of such systems are cavity magnonics[9] where microwave photons of a specific frequency are strongly coupled to a magnonic system such as a yttrium iron garnet sphere, or synthetic antiferromagnets (SAFs)[10,11] where Ruderman–Kittel–Kasuya–Yosida (RKKY) exchange-coupled ferromagnetic multilayers give strong magnon–magnon interactions on the order of GHz[9–11] but may only assume a handful of states (e.g., positively or negatively saturated) due to the constraints of strong exchange coupling.

Artificial spin systems such as artificial spin ice (ASI)[2,4,5,12] rely on relatively weak, longer-range dipolar coupling for information transfer between discrete islands rather than the strong, short-range exchange coupling used in SAFs. Impressive magnonic functionality has been demonstrated in ASI[13–20] including magnon mode hybridisation[19,21] and nonlinear multi-magnon scattering[20], but typically at smaller magnitudes rather than the many-GHz scale effects in SAFs[19,21]. To advance functional magnonic systems, solutions must be found which incorporate both vast reconfigurability and high-intensity magnon coupling phenomena.

One promising direction is the growing range of 3D nanomagnetic systems[22,23]. Recent technological advances enable sophisticated fabrication and characterisation of 3D magnetic nanostructures, but questions remain on how to precisely control such systems. Artificial spin systems have made impressive forays into the third dimension, led by the likes of Ladak et al.[24–29], Donnelly et al.[30–34], and Fernández-Pacheco et al.[35,36] alongside others[37–39], though these remain exchange-coupled rather than dipolar. Three-dimensional magnonic crystals have also been explored, with excellent studies by Gubbiotti et al.[40,41] and Barman et al.[28,29] amongst others[42–44].

Three-dimensional nanomagnetic systems offer greatly increased freedom relative to conventional two-dimensional approaches in terms of expanding and exploring the range of accessible magnetic states. The ability to finely engineer the 3D spatial position of dipolar charges offers an attractive route to engineering stronger effective dipolar coupling. However, these increased system design freedoms are accompanied by a requirement of far more challenging nanofabrication and measurement approaches. Three-dimensional systems typically demand complex patterning techniques such as two-photon lithography[26,27] or focused electron beam-induced deposition[23,35], and specialist measurement methods such as X-ray tomography[34]. These issues serve to restrict progress on studying three-dimensional magnetic systems and the new physics they offer.

Here, we present a three-dimensional nanomagnetic architecture enabling both high microstate reconfigurability and strong dynamic coupling, while requiring only simple, widely available fabrication and measurement techniques. The system is a multilayered 3D artificial spin system comprising two magnetic layers (NiFe), separated in $\hat{z}$ by a non-magnetic spacer layer (Al). Dynamic and static dipolar coupling between the magnetic layers is strong due to the relatively thin non-magnetic spacer layer (35 nm) and large degree of stray dipolar field emanating from nanoislands relative to larger structures or thin films. The multilayered system architecture allows us to reach an extremely strong dipolar coupling regime giving rise to a host of complex emergent magnetic dynamics including ultrastrong magnon–magnon coupling, GHz-scale mode frequency shifts and chiral spin-texture control.

While limited in system design freedom relative to 'true' 3D lithography approaches[26,27,33,35], the system microstate space is greatly expanded relative to conventional 2D nanomagnetic structures. Each nanoisland layer may assume four distinct states (two macrospin and two vortex, termed 'artificial spin-vortex ice'[45]) and the state of each magnetic layer may be independently addressed via global magnetic field as the layers have different thicknesses (20 and 30 nm, respectively) giving a coercive field offset. This gives 16 states per 3D nanoisland and grants a vastly reconfigurable $16^N$ microstate space.

A 3D lateral inter-layer offset is introduced via shadow deposition to break dipolar coupling symmetry between chiral vortex states, which we exploit for chirality-selective magnetic vortex state programming and resultant magnonic spectral control.

The strong inter-layer dipolar coupling is both static and dynamic. Static components are leveraged for GHz-scale mode frequency shifts between microstates and for vortex state chirality control. Dynamic components enable ultrastrong magnon–magnon coupling between magnons in the upper and lower magnetic layers, leading to mode hybridisation and an anticrossing gap with normalised coupling rate $\frac{\Delta f}{\nu} = 0.57$.

## Results
### 3D magnonic metamaterial architecture
The system considered here is a nanopatterned square ASI array of stadium-shaped 3D nanoislands comprising four distinct layers. From substrate (SiO$_2$) to top (Fig. 1a): NiFe (30 nm)/Al (35 nm)/NiFe (20 nm)/Al (5 nm). The state of each magnetic layer is independently programmable, with 'hard' (30 nm NiFe, lower layer) and 'soft' (20 nm NiFe, upper layer) layers switching at higher and lower relative $H$ fields, respectively. Islands are lithographically defined via electron beam lithography and thermal evaporation liftoff process with dimensions of $550 \times 140 \times 90$ nm, with vertex spacing of 125 nm, measured island-end to vertex-centre. A 50 nm lateral displacement in the $\hat{y}$ direction (Fig. 1b) between hard and soft NiFe layers induced via shadow deposition[46] for vortex chirality control.

An attractive feature of this design is with a single lithography step, an arbitrary number of layers can be deposited without breaking vacuum.

Each magnetic nanoisland layer can assume four distinct magnetisation states, 'up' or 'down' macrospins and clockwise (CW) and anticlockwise (ACW) vortices[45] (Fig. 1c). The combination of four states per layer and the two magnetic layers leads to 16 distinct states per 3D nanoisland, shown in Fig. 1d–g. This gives a vast $16^N$ microstate space in an $N$-island system. Crucially, each of the 16 nanoisland states exhibits distinct magnon dynamics, creating a deeply programmable magnonic metamaterial.

Figure 1h shows examples of the rich mixed microstates possible in this system, highlighting the difference in stray dipolar field strength between parallel (strong contrast) and antiparallel (weak contrast) macrospin states. The all-antiparallel state (Fig. 1i) exhibits a high number of high-energy three-in (aka 'type 3 monopole'[1]) and four-in ('type 4 monopole') vertex configurations. This occurs as the antiparallel state reduces the stray dipolar field amplitude at vertices by closing flux between layers, leading to a reconfigurable deactivation of the ice-rules governing spin system frustration – discussed further in Supplementary Note 1.

The fabrication parameters are tuned to give the strongest inter-layer dipolar coupling possible while keeping all states stable in zero applied magnetic field ($H_{ext} = 0$). Dipolar coupling strength is parameterised by the normalised magnon–magnon coupling rate $\frac{\Delta f}{\nu}$ described below in the discussion of Fig. 3. For thinner non-magnetic spacer layers or lower nanoisland length/width aspect ratios, the parallel macrospin state becomes unstable in low/zero field, with the dipolar field emanating from the hard-layer overcoming the coercivity of the soft-layer and spontaneously switching into an antiparallel macrospin state. The island aspect ratio is tuned to ensure the bistability of macrospin and vortex states to maximise state reconfigurability[45].

An appealing feature of this system architecture is its compatibility with commonly used experimental measurement techniques. Often three-dimensional systems require specialist techniques and

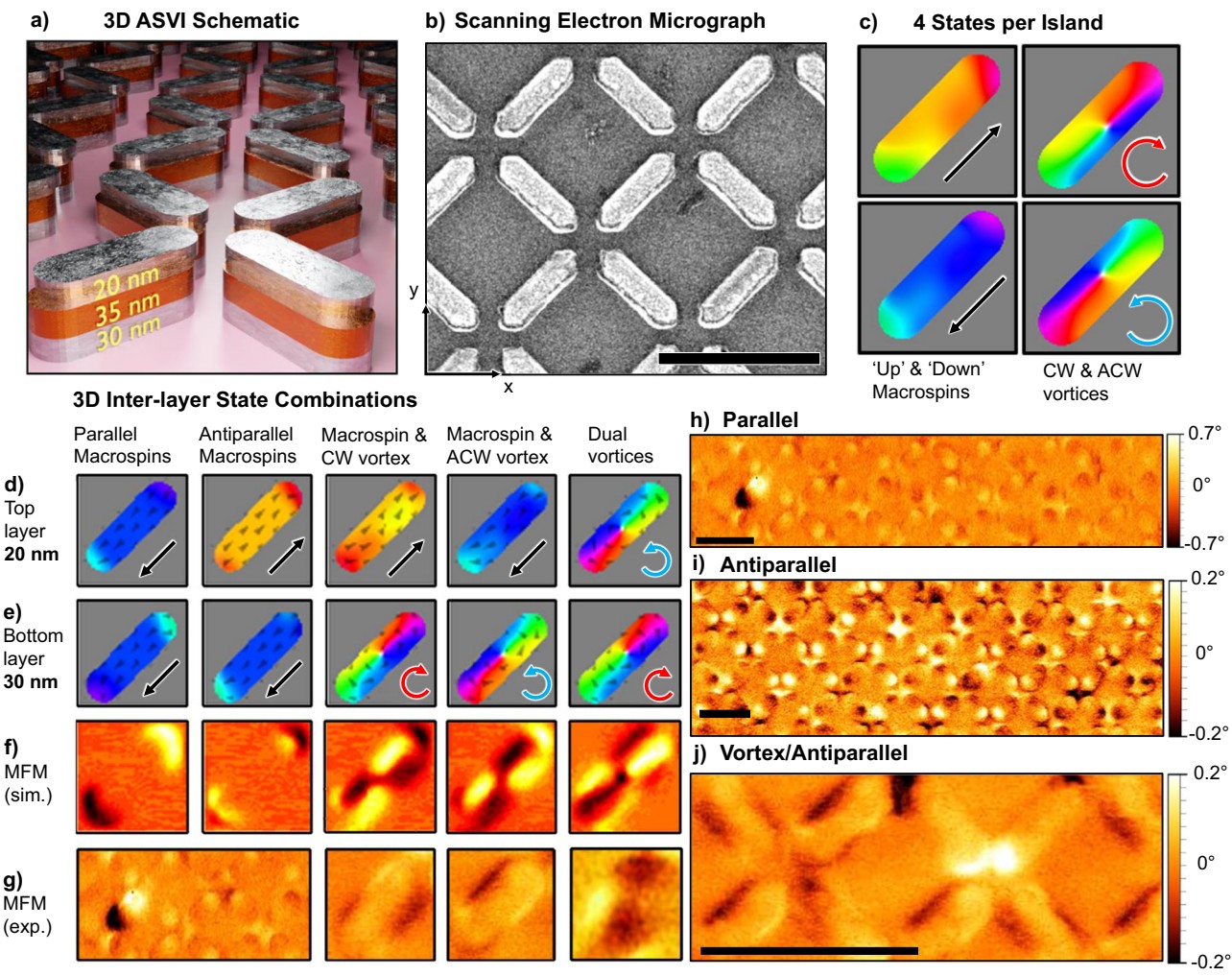

**Fig. 1 | 3D multilayered artificial spin-vortex ice. a** Schematic rendering of array architecture, showing top 'soft' magnetic layer (20 nm NiFe), non-magnetic spacer (35 nm Al) & bottom 'hard' magnetic layer (30 nm NiFe). Nanoislands are 550 nm long, 140 nm wide. Inter-island spacing is 125 nm, measured island-end to vertex-centre. **b** Scanning electron micrograph of the system, 1 μm scale bar. Relative $\hat{x}$ and $\hat{y}$ directions are shown. **c** Micromagnetic simulations (MuMax3) of the four possible magnetisation states per layer, two macrospin & two vortex. **d**–**g** 3D multilayer nanoisland magnetisation states. Micromagnetic simulations of top (**d**) and bottom (**e**) layer nanoislands are shown alongside simulated (**f**) and experimental (**g**) MFM contrast taken at 50 nm above the array. Experimental MFM of a state containing both parallel and antiparallel macrospin states is shown in the bottom left, demonstrating the large difference in stray dipolar field between parallel (strong contrast, high stray field) and antiparallel (weak contrast, low stray field) macrospin states. Experimental MFM of mixed microstates, showing (**h**) a single parallel macrospin (bottom left, strong contrast) against many antiparallel macrospins (weak contrast) antiparallel macrospin and macrospin/ACW vortex states, (**i**) a field-demagnetised antiparallel macrospin state showing many traditionally energetically unfavourable 'type 3' and 'type 4' ASI vertices and (**j**) a single parallel macrospin (bottom left, strong contrast) against many antiparallel macrospins (weak contrast). 1 μm scale bars.

equipment to resolve states. Here, conventional magnetic force microscopy (MFM) can distinguish all 16 inter-layer nanoisland states with examples of simulated (MuMax3[47]) and experimental MFM given in Fig. 1f and experimental MFM of larger mixed microstates in Fig. 1g.

## Microstate switching and dynamics

The system exhibits a three-stage magnetic reversal process, shown by the magneto-optical Kerr effect (MOKE) measurement in Fig. 2a. Beginning from a negatively saturated parallel macrospin state ($M_{P-}$) at −50 mT and sweeping field in the positive $\hat{x}$ direction, the system switches into an antiparallel macrospin state ($M_{AP}$) from 5 to 10 mT with hard-layer magnetised negative, soft-layer magnetised positive. The system then switches from 26 to 30 mT into a 3D specific state without a 2D nanoisland analogue, a parallel macrospin state with both layers positively magnetised but with a substantially edge-curved state where the magnetisation at the end of the nanoisland ends becomes locked in a highly curved exaggerated S- or C-state[48,49] due to the influence of strong dipolar field emanating from the other 3D magnetic

layer ($M_{P+}^{*}$). The edge magnetisation in the top and bottom layers is non-collinear and is only achieved when the previous state consists of antiparallel macrospins. This state is resolved by the lower relative MOKE signal (0.78) and has a unique ferromagnetic resonance (FMR) from the edge-curvature, discussed further below and shown in Supplementary Fig. 2. At 43 mT, the edge-curvature straightens out into saturated positively magnetised parallel macrospin state ($M_{P+}$) and the edge magnetations are now collinear. The presence of vortices is not seen in the hysteresis loop, as they are gradually nucleated over multiple field loops[45].

Figure 2b shows an FMR field vs. frequency plot where $\boldsymbol{H}_{ext}$ is continuously applied while recording FMR spectra. This leads to changes in magnon mode dynamics both from microstate shifts and from increasing applied field $\boldsymbol{H}_{ext}$ amplitude. The relative magnetisation direction of the magnetic states and textures hosting the various magnon modes may be discerned from the mode-frequency gradient with field. Positive mode gradient corresponds to modes aligned with the applied field, and vice-versa for negative gradient. This can be seen

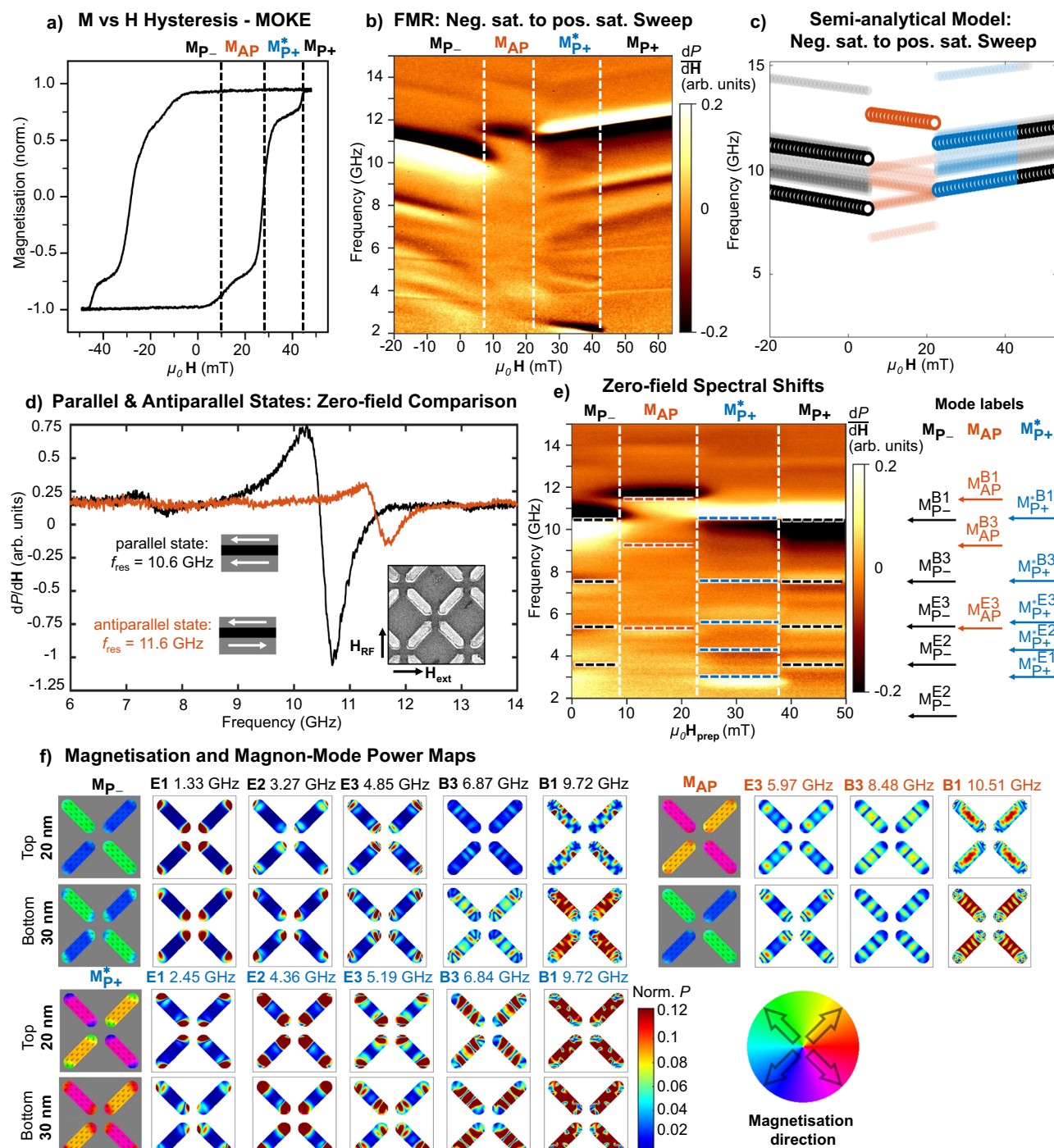

**Fig. 2 | Static and dynamic magnetic response. a** $M$ vs. $H$ hysteresis loop via magneto-optical Kerr Effect. Central switching fields are indicated by dashed vertical lines. Parallel to antiparallel transition ($M_{P-} \rightarrow M_{AP}$) occurs between 5 and 14 mT, antiparallel-to-parallel with curved edges transition ($M_{AP} \rightarrow M_{P+}^*$) occurs between 20 and 26 mT. Substantial nanoisland end magnetisation curvature persists up to 43 mT, above which the magnetisation assumes a stable straightened parallel macrospin state ($M_{P+}$). **b** FMR field vs. frequency sweep ($M_{P-} \rightarrow M_{AP} \rightarrow M_{P+}$). Three distinct switching events are observed matching the hysteresis loop in (**a**) including edge curvature straightening. **c** Semi-analytical model calculation of mode frequencies vs. field for the same negative saturation to positive saturation field sweep as performed in (**b**). Marker colour code refers to the same parallel/antiparallel states as in (**b**). **d** Ferromagnetic resonance (FMR) spectra for parallel

(black) and antiparallel (red) macrospin states, showing a 1 GHz frequency shift between the states at $H_{ext} = 0$. SEM schematic defines DC external field $H_{ext}$ and RF field $H_{RF}$ orientations for FMR data panels in this figure. **e** Remanence FMR sweep from negative to positive saturation. $X$-axis denotes a 'preparation field' which is applied before returning to zero field to measure the remanence spectra. The data show the same three switching events as (**b**), without the convolution of increasing applied field changing the Kittel frequency of the magnon modes consequently all frequency shifts are only due to dipolar field landscapes in the 3D nanoarray. **f** MuMax3 simulated spatial magnetisation (grey background:colour wheel) and magnon power maps (white background:colour bar) at $H_{ext} = 0$ (note: simulated frequencies lower compared with experiment).

in the negative gradient 2 and 5 GHz edge-curvature modes between 22 and 43 mT. Here, the central region of the magnetic texture behaves as a macrospin aligned with the applied field (11–12 GHz, 8–9 GHz and 6–7 GHz modes) and the curved edge regions (2 and 5 GHz) behave as magnon edge-modes[5,13,15,49,50] anti-aligned with the applied field.

To model the system dynamics, we use the semi-analytical code 'Gænice'[51] that allows computationally efficient calculation of the total dipole field of a finite-sized ASI array of arbitrary configuration (details in 'Methods'). As systems expand into 3D and more complex architectures, micromagnetic simulation approaches become computationally expensive and alternative modelling approaches are attractive. Figure 2c shows Gænice calculation of field-swept magnon mode frequencies, following the FMR sweep protocol in Fig. 2b. Good correspondence of mode structure across the different parallel and antiparallel states is observed, including the prediction of an opposite-gradient lower-frequency mode in the antiparallel state (red points), examined further in the next section. In addition, the low-frequency mode detected in the $M_{P+}^{*}$ region is not recovered with Gænice. This implies that such a mode results from a nonlinear magnetisation texture, requiring further analysis with micromagnetic simulations, presented later as a vortex mode.

The variation in local static dipolar field configuration between island states leads to large GHz-scale magnon frequency shifts. Crucially, as the dipolar field responsible for the frequency shifts is supplied passively by the nanomagnets themselves, these GHz shifts are available at $H_{ext} = 0$. This is attractive from a technological standpoint, allowing large mode shifts without the energy and engineering constraints of continuously applying field.

Figure 2d shows $H_{ext} = 0$ FMR frequency sweeps taken in a pure parallel macrospin state $M_P$ (black curve, $f_{res} = 10.6$ GHz) and pure antiparallel macrospin state $M_{AP}$ (red curve, $f_{res} = 11.6$ GHz), a $\Delta f = 1$ GHz zero-field frequency shift – substantially higher than typically observed in magnonic crystal systems[18,21,52].

This frequency shift arises from the static component of the stray dipolar field. We can compare the magnitude of frequency shift to the magnitude of the static dipolar field felt by the magnetic layers. Micromagnetic simulations of the spatially distributed dipolar field magnitude give a difference in static dipolar field between parallel and antiparallel macrospin states of 33–38 mT (shown in Supplementary Figs. 3 and 4). Comparing this with the field-swept FMR data in Fig. 2b and using a first-order approximation of a linear field/frequency gradient for the main Kittel mode (10–12 GHz), we see that 33 mT of applied $H_{ext}$ field is required to generate a mode shift of 1 GHz – a good correspondence with the simulated dipolar field magnitude.

These zero-field mode frequency shifts can be programmed on a per-island basis in mixed array states (Fig. 1h–j). States may be locally written via surface probe or optical writing techniques[53–56], giving a strongly programmable GHz meta-surface.

To demonstrate the degree of zero-field magnon configurability, Fig. 2e shows a 'remanence FMR' sweep[57] where all spectra are recorded at $H_{ext} = 0$. The x-axis represents a 'state preparation field' $H_{prep}$, which is applied momentarily prior to spectra measurement.

A negatively saturated parallel macrospin state is initially prepared and $H_{prep}$ field is swept in a positive direction. This remanence approach allows the magnon mode-frequency shifts arising purely from the array microstate to be deconvoluted from the $f_{res}$ shifts due to changing $H_{ext}$ amplitudes.

In the parallel state (0–8 mT) a main dominant mode frequency is observed at 10.6 GHz ($M_{P-}$ in Fig. 2b) with corresponding micromagnetic simulated spatial magnetisation and power maps (MuMax3) shown in Fig. 2f. The antiparallel state (8–26 mT) shows the dominant mode $M_{AP}$ being blue shifted to 11.6 GHz via the stronger internal field in the antiparallel state caused by the inter-layer static dipolar field summing with the nanoisland magnetisation, depicted schematically in Supplementary Fig. 5.

Between 26 and 40 mT the system enters the edge-curved parallel macrospin state $M_{P+}^{*}$ and the main Kittel mode redshifts back to 10.6 GHz. At lower frequency the strong edge-curvature results in a pronounced edge mode at 2.6 GHz (labelled E1, simulated spatial power maps in Fig. 2f), higher intensity than in typical single-layer magnonic crystals due to strongly curved edges induced by the inter-layer coupling.

Detecting edge modes in nanostructures can be challenging relative to main Kittel modes due to their small volume fraction and sensitivity to nanofabrication imperfections. The strong 3D dipolar coupling in this system increases the volume of the curved edge regions in the parallel macrospin state and hence renders them more easily resolvable relative to a single-layer system, with the 2.6 GHz edge mode in this state having equal amplitude to the main Kittel mode at 10.6 GHz.

In the antiparallel state, edge magnetisation no longer curves into S- or C-shaped states[49]. This is due to the strong dipolar attraction between the edge magnetisation of one nanoisland layer and the edge magnetisation of the adjacent layer. This attraction leads to a straightened-out magnetisation texture at the nanoisland edges, whereas the repulsion in the parallel macrospin state leads to curved edge states. This is reflected by the absence of detectable lower-frequency edge modes in the antiparallel state, seen in Fig. 2b, e and f. Further discussion and data on the reconfigurable edge-curvature modes are given in Supplementary Fig. 2.

Above 41 mT the system saturates into a positively magnetised parallel macrospin state and the enhanced edge-curvature is straightened out. The system is now in an identical state to its initial zero-field conditions, albeit positively magnetised rather than negatively.

## Ultrastrong magnon–magnon coupling

The strong static dipolar coupling between nanoisland layers leads to the rich microstate pathways, diverse magnon modes and large mode-frequency shifts discussed above. The dynamic component of the inter-layer dipolar field drives high-intensity coupling between spatially separated magnon modes located in the upper and lower magnetic layers, leading to resultant inter-layer magnon mode hybridisation.

Here, we demonstrate ultrastrong magnon–magnon coupling between magnons in the upper and lower layers, generated using only via dipolar interaction with no inter-layer exchange coupling present. The dipolar interaction, traditionally considered relatively weak, is leveraged for these high-intensity effects via the small separation between magnetic poles made possible by our multilayer 3D architecture, and by patterning a dense nanoisland array which has a far higher degree of stray dipolar field than a thin-film system.

Figure 2f shows MuMax3 simulated magnetisation and spatial magnon mode power maps for the edge-modes (E) and bulk Kittel modes (B) in the parallel (black), antiparallel (orange) and edge-curved parallel (blue) states. The key difference between the $M_{P-}$ and $M_{P+}^{*}$ states is the configuration of the nanoisland edge magnetisation. After passing through the $M_{AP}$ state the $M_{P+}^{*}$ state retains an antiparallel configuration of the nanoisland ends, resulting in the same blue shifting of the edge states. The E1 mode in particular is at too low a frequency to be detected for $M_{P-}$ but is within the experimentally measured range for $M_{P+}^{*}$.

Figure 3a shows a micromagnetic simulation of a trilayer system (two upper and lower magnetic layers, interstitial non-magnetic spacer layer) in the antiparallel macrospin state with a 250 nm non-magnetic spacer. Here, magnon modes in the upper (purple) and lower (green) layers are spatially well separated and non-interacting, evidenced by the modes crossing through each other with no gap. Figure 3b–d shows simulations with decreasing non-magnetic spacer thicknesses (80, 40 and 20 nm, respectively). As the spacer thickness is reduced and the dynamic component of the inter-layer dipolar field becomes

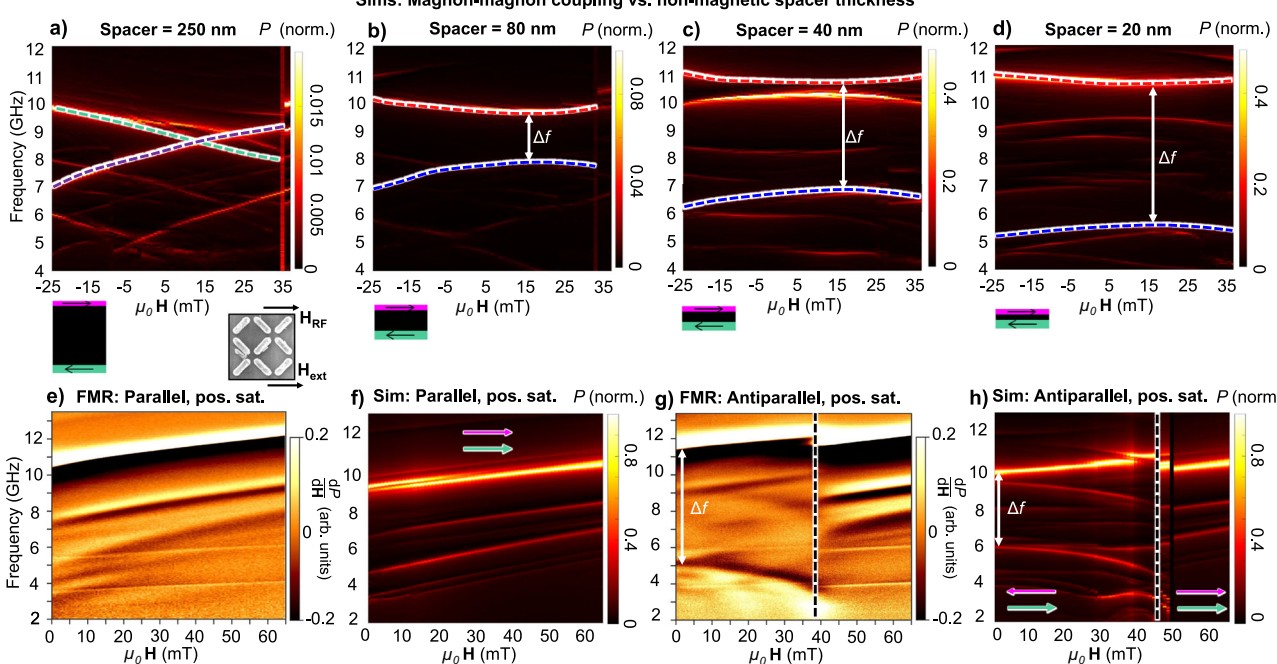

**Fig. 3 | Ultrastrong magnon–magnon coupling. a–d** Simulations of anticrossing gap opening and mode hybridisation forming acoustic and optical modes as the inter-layer spacer thickness is varied from 250 to 20 nm illustrated by purple and green schematics (to scale). SEM schematic defines DC external field $H_{ext}$ and RF field $H_{RF}$ orientations for FMR data panels in this figure. For a 250 nm spacer layer (**a**) each mode corresponds to a single magnetic layer, no hybridisation occurs and modes cross. For thinner spacer layers (**b**, **c**) modes are hybridised with both magnetic layers oscillating together. In-phase (upper-frequency branch, acoustic mode) and out-of-phase (lower-frequency branch, optical mode) modes are separated by an anticrossing gap of width Δ*f*. **e** Positive field direction FMR sweep on a positively saturated parallel macrospin state. Array remains in a parallel macrospin state throughout sweep with no optical mode observed. In-phase dominant and subharmonic modes are observed. **f** Corresponding micromagnetic simulation to (**e**) reproducing dominant in-phase magnon mode and subharmonics. **g** Positive field direction FMR sweep on an antiparallel macrospin state with hard-layer positively magnetised, soft-layer negative. A switch from antiparallel-to-parallel macrospin state is observed at 40–45 mT. An optical mode is observed at 4.96 GHz, with the acoustic mode at 11.51 GHz and a 6.55 GHz gap. **h** Corresponding micromagnetic simulation to (**g**). Thick and thin white arrows refer to the magnetisation of hard and soft layers, respectively. Switch from antiparallel-to-parallel state is observed from 45 to 50 mT and marked by white dotted vertical line.

larger, an anticrossing gap Δ*f* opens between the modes. This occurs as the dipolar magnon–magnon coupling between magnons in the upper and lower layer becomes sufficiently strong to drive mode hybridisation, generating distinct acoustic/in-phase (red mode, higher frequency) and optical/out-of-phase (blue mode, lower frequency) mode branches where magnons in both layers hybridise and oscillate together. At spacer thickness below 40 nm, additional higher-order modes appear in the spectra. Spatial power and phase maps of these modes are shown in Supplementary Fig. 7.

Moving to experimental data, Fig. 3e shows FMR spectra of the array prepared in a positively saturated parallel macrospin state (+50 mT field), then swept from 0 to 65 mT in a positive field direction. Spectra are recorded with the RF microwave field $H_{RF}$ and the globally applied field $H_{ext}$ in a parallel geometry, illustrated by the SEM image in Fig. 3. Figure 3f shows the corresponding micromagnetic simulation. Five modes are observed, corresponding to various resonances of the parallel macrospin state. All modes exhibit a positive frequency/field gradient.

Figure 3g shows experimental FMR spectra of an antiparallel macrospin state. Field is swept up until the array switches to a positively magnetised parallel macrospin state at 43 mT. Corresponding simulation in Fig. 3h. As seen in the simulations in Fig. 3b–d a new lower-frequency mode with negative gradient is observed in the antiparallel spectra, separated from the higher frequency mode by 6.55 GHz at 0 field.

Examining the character of the lower-frequency mode in micromagnetic simulation, Fig. 4a shows the power, $\hat{y}$-component phase maps and layer-separated time-evolution traces for the high (11.6 GHz)

and low (4.96 GHz) frequency modes for a 30 nm spacer at 0 field. Due to the shape anisotropy of the nanoislands, the precession amplitude is most significant in-plane so we examine spatial phase maps generated from the $\hat{y}$-magnetisation component. We also generate time-evolution traces of the $\hat{y}$-magnetisation for the top and bottom magnetic layers, averaged over an entire nanoisland. Full phase maps for the 30 and 250 nm spacer cases are shown in Supplementary Fig. 6. The high-frequency mode is seen to have an in-phase (acoustic) relationship between magnons in the upper and lower magnetic layers, while the lower-frequency mode shows layers oscillating 180° out-of-phase (optical). This further confirms the acoustic and optical character of the hybridised magnon modes, showing that the dipolar inter-layer magnon–magnon coupling between magnons in the upper and lower magnetic layers is sufficiently strong to generate mode hybridisation.

In previous studies by some of the authors[19,21] the coupling is mediated by nanoislands placed end-to-end in a two-dimensional array and exhibits an exchange-like energy scaling where the optical mode has higher energy than the acoustic. When the nanoislands are placed on top of each other in the three-dimensional geometry here, both nanoisland ends are involved in the coupling and the energy scaling is dipolar in nature so the acoustic is now higher energy (see Supplementary Note 4 for more detailed explanation). An analogue is observed in two-nanoisland plasmonic systems, with thorough investigations of island geometry and relative mode frequency showing the same behaviour[58].

Further evidence of the acoustic and optical character of the modes is provided by the fact that the optical mode is only clearly

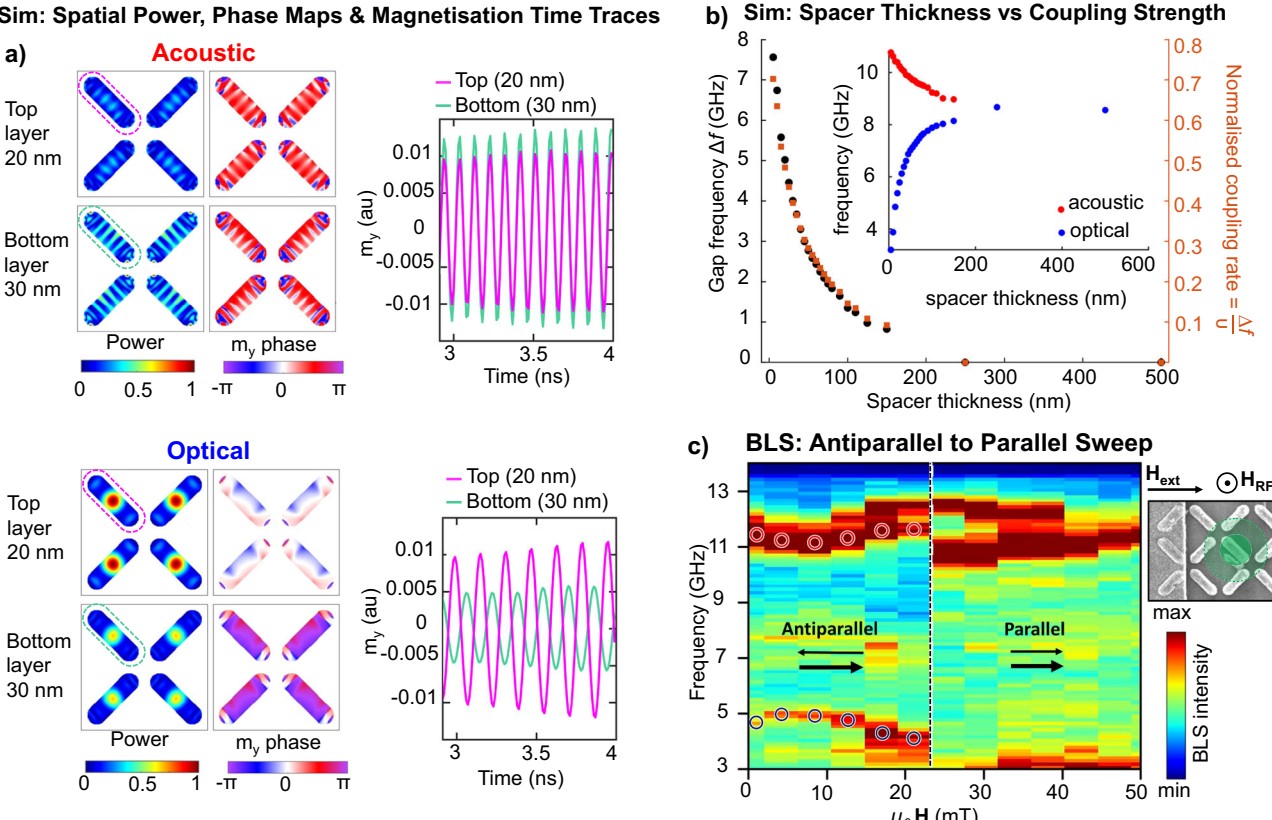

**Fig. 4 | Further evidence of magnon–magnon coupling. a** Power and phase plots of the acoustic (10–11 GHz) and optical (5–6 GHz) modes for the top and bottom magnetic layers are shown, with in-phase and out-of-phase inter-layer relationships observed for the acoustic and optical modes, respectively. Time traces are reduced magnetisation of the top left nanoisland for the top (purple) and bottom (green) layers when excited with a sinusoidal excitation at the mode frequency. **b** Micromagnetic simulation of acoustic/optical mode frequency gap (left-axis) and normalised coupling rate (right-axis) of the upper and lower layer magnons as a function of spacer thickness. Optical and acoustic mode frequencies (blue and red points, respectively) are plotted in inset. **c** Micro-focused BLS spectra taken from a

500 nm laser spot. RF excitation field is supplied by on-chip microfabricated CPW, in the $\hat{z}$-direction as nanostructures are situated between CPW ground and signal line. Field sweeps from antiparallel (0–25 mT) to parallel (25–50 mT) macrospin state. Acoustic (11.5 GHz) and optical (5 GHz) modes are observed in the antiparallel state. Lower antiparallel-to-parallel switching field is observed than in Fig. 3g as the nanoislands fabricated for the BLS sample are 170 nm wide, vs. 140 nm for the FMR sample. Inset shows SEM image of BLS sample and on-chip CPW (left edge). Green dot indicates size and position of ~500 nm BLS laser spot for the spectra in (**c**) with fainter green circle indicating radius of potential spot drift over the measurement.

detected when $H_{RF}$ and $H_{ext}$ are in a parallel geometry – using a conventional perpendicular FMR geometry between the two fields results in only the higher frequency acoustic mode being clearly resolved (Supplementary Fig. 8e) which matches the behaviour observed in SAFs[11]. This occurs as a perpendicular RF field cannot generate the requisite opposite torque on the two magnetic layers required to excite out-of-phase optical modes.

A quantitative assessment of magnon–magnon coupling strength can be expressed via a figure of merit termed the normalised coupling rate[59,60], the ratio of the gap width and upper mode frequency $\frac{\Delta f}{\nu}$ where $\Delta f$ is the width of the frequency gap between acoustic and optical modes at its narrowest point (here $\Delta f = 6.55$ GHz at $H_{ext} = 0$) and $\nu$ the frequency of the upper mode (acoustic) at the same point. Here $\nu = 11.51$ GHz, giving a normalised coupling rate of $\frac{\Delta f}{\nu} = 0.57$ between magnons in the upper and lower layers. It should be noted that the normalised coupling rate quantifies the strength of magnon–magnon coupling between the magnons in the upper and lower nanoisland layers. This coupling hybridises the upper and lower layer magnon modes, generating the acoustic and optical modes. It is not the coupling strength between the acoustic and optical modes.

For $\frac{\Delta f}{\nu}$ values above 0.1, a system can be said to be in an 'ultrastrong coupling' regime[59,60]. With the ultrastrong coupling we describe here, we do not invoke the picture used in some light-matter coupling systems where standard approximations break down and unconserved

virtual excitations become active[59]. Instead, we use the ultrastrong regime of the normalised coupling rate as a quantitative assessment of the high intensity of dynamic coupling phenomena active in this system, demonstrating of the capacity for dynamic dipolar interactions to generate emergent many-body coupling phenomena.

The $\nu = 11.51$ GHz value here differs slightly from the 11.60 GHz value in Fig. 2b, c and d, here the RF field is applied parallel to $H_{ext}$ in order to access a parametric pumping geometry and couple to the optical mode[11] while in Fig. 2 the RF field $H_{RF}$ is applied perpendicular to $H_{ext}$ to exert maximal torque on the magnetisation. This difference in relative field geometries may account for the observed 90 MHz frequency difference between Figs. 2 and 3.

Figure 4b shows micromagnetic simulations of how the gap frequency $\Delta f$, acoustic and optical mode frequencies and the normalised coupling rate $\frac{\Delta f}{\nu}$ vary as a function of the non-magnetic spacer layer thickness, with gaps of 7.5 GHz available. For a 35 nm spacer thickness matching experiment, simulations show a normalised coupling rate of $\frac{\Delta f}{\nu} = 0.37$, lower than experiment but still in the ultrastrong coupling regime. Simulations suggest high coupling rates up to 0.63–0.7 are available for spacer thicknesses of 10–5 nm, with potentially higher values in experimental systems if the relative experiment/simulation relationship holds. Between 150 and 250 nm spacer thickness, the inter-layer dynamic dipolar coupling becomes too weak to generate hybridisation between magnons in the upper and lower magnetic

layers. In this regime it is reasonable to say that there is negligible magnon–magnon coupling.

So far, our experimental spin-wave spectra have been provided by 'flip-chip' FMR using mm-scale arrays comprising ~$10^8$ nanoislands. To examine single nanoisland behaviour and demonstrate the local programmability of our 3D magnonic metamaterial, we optically probe the integrated dynamic magnon response from a few nanoislands (~500 nm active area) via micro-focused Brillouin light scattering (BLS, details in 'Methods'). Figure 4c shows BLS spectra taken using a ~500 nm spot laser, with an antiparallel-to-parallel macrospin state trajectory corresponding to Fig. 3g, h. Magnon dynamics are excited by an on-chip microfabricated coplanar waveguide (CPW) with the metamaterial array positioned between the ground and signal lines of the CPW to give a $\hat{z}$-direction RF field, shown via SEM in Fig. 3c inset with BLS laser spot size and position indicated in green. The array fabricated for this on-chip CPW BLS sample has slightly different lateral nanoisland dimensions, $550 \times 170$ nm, with the same layer thicknesses as discussed previously. These slightly wider islands have lower switching fields than the $550 \times 140$ nm islands used for FMR experiments but otherwise function similarly.

The micro-BLS spectrum shows acoustic and optical modes ($f_{acoustic} = 11.5$ GHz, $f_{optical} = 4.8$ GHz at $H_{ext} = 0$) in the antiparallel state. Here, rather than the distribution of switching fields seen in FMR arising from a huge array switching island-by-island, an abrupt single antiparallel-to-parallel switching event is seen at 25 mT. The low-frequency optical mode now abruptly deactivates, following behaviour seen in FMR and micromagnetic simulation. This few-island data demonstrate the degree of local reconfigurability in this metamaterial and the rich dynamics active in discrete nanostructures enabled by the 3D nanomagnetic architecture. Interestingly, a $\hat{z}$ RF field is not capable of coupling to optical modes in multilayered thin-film geometries such as SAFs as it fails to satisfy the required geometric conditions[11]. This shows our system architecture is capable of expanding the range of allowed RF coupling geometries. This occurs by both introducing substantial 3D edge-curvature and canting to the nanoisland magnetisation, breaking in-plane geometric orthogonality, and employing different thickness magnetic layers, breaking $\hat{z}$ symmetry. RF coupling geometry is examined further in Supplementary Fig. 8.

The BLS data show a minimum anticrossing gap at a small finite positive field of 7 mT. This matches the behaviour seen in our micromagnetic simulations and occurs as the two magnetic layers have differing thicknesses (equal thickness layers give a zero-field minimum gap). This is clearer in the BLS data as it measures just a few islands, while the FMR data measures a $10^8$ nanoisland array where nano-patterning imperfections (often termed 'quenched disorder') can average out such effects. The mode seen above the main Kittel mode in the BLS data is a consequence of the BLS being measured using a $\hat{z}$ RF field, with corresponding $\hat{z}$ micromagnetic simulations showing similar behaviour (Supplementary Fig. 8).

### Chiral symmetry breaking of magnetic vortex states

Here, we demonstrate selective control of preparing CW or ACW vortex states in one magnetic layer via the static dipolar field of macrospin states in the adjacent layer. This allows programmable chiral control. By placing one magnetic layer in a macrospin state, we may exploit its dipolar field to selectively control the chiral CW/ACW state of magnetic vortices in the adjacent layer. This chiral selectivity may be activated or deactivated on-demand by programming the state of the macrospin layer, allowing three regimes: forced CW vortices, forced ACW vortices and stochastic mixed CW/ACW vortices. Demonstration of this chiral symmetry breaking and its application for microstate control are shown in Fig. 5.

The inter-layer $\hat{y}$ offset mentioned previously now becomes crucial. Figure 5a, b compares two cases: applying $H_{ext}$ perpendicular to the offset direction (Fig. 5a) which breaks chiral symmetry, and

applying $H_{ext}$ parallel to the offset (Fig. 5b) which maintains chiral symmetry. Introducing a relatively small 50 nm lateral offset to the system architecture enables full chirality control.

Figure 5a shows the energy difference between CW and ACW vortices normalised by the CW vortex energy $\frac{E_{CW} - E_{ACW}}{E_{CW}}$ (bottom), where the energy is a sum of the exchange and demagnetisation energies in both magnetic layers. Energies are plotted as a function of $H_{ext}$ ($x$-axis) and inter-layer offset width ($\hat{y}$-axis). Here vortices are present in the 30 nm layer with an $-\hat{x}$-magnetised macrospin state in the adjacent 20 nm layer.

A strong chiral energy degeneracy lifting is observed, with an energy difference of up to 8% of the total state energy. The mechanism is described in Supplementary Fig. 10. Degeneracy lifting is observed at both $H_{ext} = 0$ and under applied field, with the strongest degeneracy lifting when $H_{ext} = 0$ is oriented parallel to the macrospin layer in the $-\hat{x}$ direction. Reversing the macrospin magnetisation to $+\hat{x}$ inverts the favoured lower-energy CW/ACW vortex state.

The full chirality of magnetic vortex state may be characterised as combination of the 'circulation' (the CW/ACW chirality of the in-plane magnetic texture) and the vortex core polarity ($\pm\hat{z}$ 'up/down' component at the centre of the vortex). The bottom two panels of Fig. 5b show that in the case of an $\hat{x}$ inter-layer offset and $\hat{x}$ $H_{ext}$, symmetry is broken for $\pm\hat{z}$ 'up/down' vortex core polarities. An energy difference up to 0.3% of the total system energy is observed, smaller than the CW/ACW energy difference which is as expected due to the vortex core comprising a much smaller volume fraction of the system relative to the in-plane magnetisation regions.

Chiral symmetry is restored when the macrospin-layer magnetisation and $H_{ext}$ are oriented along the 'Symmetric' axis, parallel to the inter-layer offset (Fig. 5b, top-left panel). Here, zero energy difference is observed between vortex chiralities at all applied fields and inter-layer offsets.

This programmable energy landscape is now leveraged for chiral microstate control. Figure 5c shows an MFM image taken after field-looping ($30 \times \pm30$ mT) along the 'broken-symmetry' axis, with the macrospin layer magnetised to lower the ACW vortex energy. The resultant microstate shows only ACW vortices present. Figure 5d shows the same process, with the macrospin layer magnetised to select for CW vortices. Here, only CW vortices are observed in the resultant microstate. Figure 5e shows an MFM image after field-looping along the symmetric axis, with no chiral degeneracy lifting and a resultant mixed CW/ACW vortex state. Figure 5f shows a larger area MFM image selecting for ACW vortices, with the dotted square corresponding to the region in Fig. 5c. Figure 5g shows MFM of a larger area mixed-chirality state, demonstrating the strong degree of chiral control and microstate selectivity enabled via our 3D metamaterial architecture. It is challenging to resolve the vortex core polarity via MFM in this multilayered system. From our simulation of the energetic degeneracy lifting between core polarity states, there is a likelihood we are controlling vortex polarity and circulation state but full confirmation of this requires further experimental work.

A number of methods exist to control the state of magnetic vortices in nanostructures, typically involving breaking the symmetry of the nanostructure itself via fabricating an asymmetrically shaped structure. Examples include disks with a sliced-off edge to give a 'D' shape[61–64], variable-width crescent-shaped rings[65], merging two disks together[66] and asymmetric bicomponent wedge rings[67]. This suite of approaches allows powerful control over CW/ACW vortex circulation and in some cases core polarity. The catch is that as the nanostructures themselves are asymmetric, different vortex states have asymmetry introduced into their dynamics and energy. This asymmetry is hard-coded at the fabrication stage and may not be deactivated or reconfigured.

A benefit of the approach described here is that by using the dipolar field of an adjacent 3D nanoisland layer to control vortex

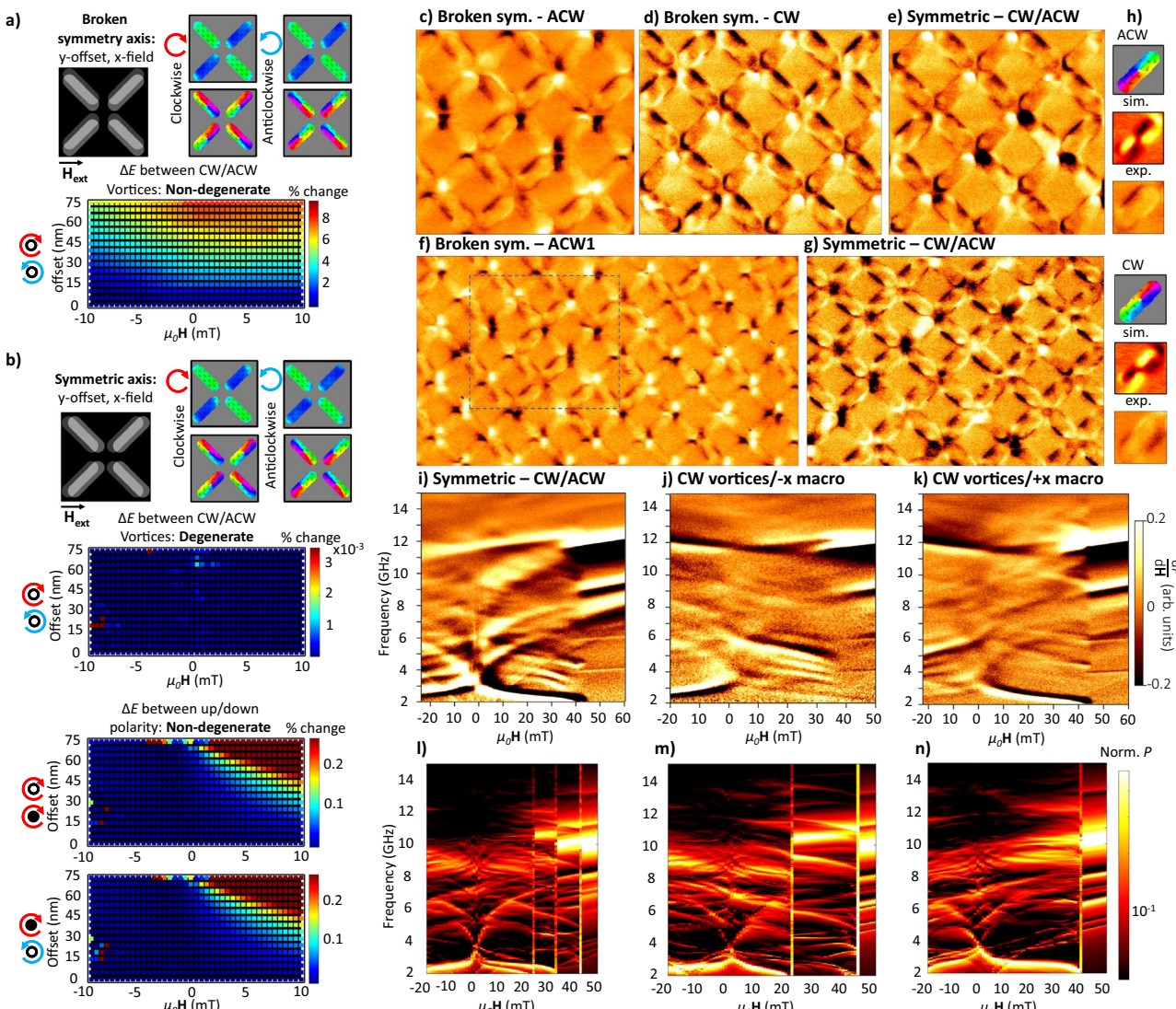

**Fig. 5 | Vortex state control. a** Chiral energy degeneracy as a function of inter-layer offset and applied field along the 'broken-symmetry' axis, perpendicular to the offset, illustrated by grey-scale inset. Energy is calculated as normalised energy difference between CW and ACW vortex states. Vortices and macrospins are in 30 and 20 nm layers, respectively. **b** Same as (**a**) but field applied along the symmetric axis, parallel to the inter-layer offset. Full data in Supplementary Fig. 12. **c** MFM image after field-looping along the 'broken-symmetry' axis. The macrospin layer is saturated to select for ACW vortices. **d** Same as (**c**) along 'broken-symmetry' axis, macrospin layer saturated to select CW vortices. **e** MFM image after field-looping along symmetric axis. Both CW and ACW vortices observed. **f** Larger area MFM of ACW state. Dotted square corresponds to the image in (**c**). **g** Larger area MFM of symmetric mixed CW/ACW state. **h** Vortex state key, showing top-to-bottom:

simulated magnetisation, simulated MFM and experimental MFM. **i** Positive field direction FMR sweep on a microstate comprising both vortex chiralities and anti-parallel macrospins with chiral edge-curvature states, field swept along symmetric axis. Macrospin magnetisation initialised in negative $\hat{y}$ direction followed by thirty ±30 mT minor loops to nucleate vortices, then FMR swept −30 to +60 mT. **j** Identical process to (**i**), but with microstate prepared and measured along 'broken-symmetry' axis. Macrospin-layer magnetisation is initially programmed in negative $\hat{x}$ direction. Chiral edge-curvature states exhibit a strongly broken spectral symmetry around $H_{ext} = 0$. **k** Identical process to (**j**), but with macrospin-layer magnetisation initialised in positive $\hat{x}$ direction. The broken spectral symmetry around $H_{ext} = 0$ is seen to have a reversed sign. (**l–n**) Simulated FMR corresponding to experimental data above.

states, the nanostructures may remain symmetric and the dipolar field may be reconfigurably programmed on a per-island basis, for instance by using local magnetic writing[53,55]. The dipolar field control may also be effectively deactivated by preparing the control layer in a vortex state that has a far lower stray dipolar field than a macrospin state.

The broken chiral symmetry may also be used as a means to further control and sculpt the spectral magnon dynamics. Figure 5i shows a mixed CW/ACW vortex microstate, prepared by negatively saturating and then applying ±30 mT minor loops along the symmetric axis to gradually nucleate vortices[45]. An extremely rich spectrum is observed, given by a combination of magnetic vortex states ($\chi$-shaped modes, 2–6 GHz). Edge-curvature states (dominant edge-curvature mode overlaps with the 2–3 GHz vortex modes, and 3 higher-frequency edge-

curvature modes 3–5 GHz) and antiparallel macrospin states (11 GHz). Examples of magnetic vortex magnon spectra without edge-curvature modes also present may be found in a prior work by some authors[45], and an annotated version of this plot is given in Supplementary Fig. 11 with modes labelled with the magnetic texture they originate from. Here the lower-frequency chiral modes have a spectral mirror symmetry around $H_{ext} = 0$, as expected for the symmetric array axis. Above 44 mT, the chiral textures all straighten out and the system assumes a positively magnetised parallel macrospin state.

Figure 5j shows a microstate prepared via an identical field-preparation sequence to Fig. 5i, but here $H_{ext}$ is applied along the 'broken-symmetry' axis and the macrospin layer is initially programmed in the negative $\hat{x}$ direction. The observed magnon spectra

are very different from the symmetric CW/ACW case. The same anti-parallel macrospin mode at high frequency is observed, but the lower-frequency chiral modes are substantially different. Here, at $H_{ext} = 0$ the edge-curvature mode abruptly jumps from 2.6 GHz in negative field to 5 GHz in positive field, giving a large 2.4 GHz mode discontinuity. This occurs as the edge-curvature magnetisation texture is strongly coupled to a vortex state in the adjacent layer. As the vortex goes from its 'favourable' (here negative field) to 'unfavourable' (positive field) $H_{ext}$ polarity, the magnetic texture is reconfigured and a corresponding jump in mode frequency is observed. This leads to a broken magnon spectral symmetry defined by the inter-layer coupling, with no such asymmetry observed when $H_{ext}$ is applied along the symmetric axis. This large magnon frequency discontinuity around $H_{ext} = 0$ without an accompanying magnetisation reversal is unprecedented in reconfigurable magnonic systems, with technological implementations including low-field frequency switching and sensing.

The directionality of this broken spectral symmetry may be programmed by the macrospin-layer magnetisation direction. Figure 5k shows the same preparation and measurement process as 5j, here the initial magnetisation of the macrospin layer is initially programmed in the positive $\hat{x}$ direction, opposite to 5j. The relative low/high frequency in positive/negative fields is reversed, demonstrating the role of the inter-layer coupling in controlling the magnon spectra field-symmetry breaking.

## Discussion

In this work, we have demonstrated that a 3D magnonic metamaterial may be engineered by vertically stacking independently programmable artificial spin systems. This architecture combines the vastly expanded range of microstates and strong dynamic phenomena enabled by a three-dimensional approach with the strong magnon interactions and inter-layer coupling more commonly found in less reconfigurable systems. The fabrication approach is relatively simple and widely available relative to 'fully 3D' approaches such as two-photon lithography or focused electron beam-induced deposition.

The host of dynamic and magnetostatic behaviours enabled by this multilayered 3D architecture is significant, from substantial dynamic phenomena including GHz mode shifts in zero field and ultrastrong magnon–magnon coupling, to chirality-selective magnetic vortex states and corresponding magnon spectra control. The magnon phase control provided by the optical and acoustic modes has implications for the growing field of coherent magnonics[68]. These wide-ranging phenomena are testament to the promise of 3D nanomagnetic systems and demonstrate the future potential once precise reconfigurable state control is mastered across 3D architectures.

In addition to the spectral and microstate control demonstrated here, the chiral ordering control may enable a host of intriguing phenomena including non-reciprocal optical dichroism[69] and chiral & non-reciprocal magnonics[70].

Future implementations of the 3D multilayered architecture have broad opportunities for advancement. Additional 3D layers may be easily introduced, with potential for diverse magnetic materials including antiferromagnets and active non-magnetic spacers enabling interfacial Dzyaloshinskii–Moriya interaction and RKKY interaction. As layer numbers increase, 3D states and textures beyond parallel/anti-parallel become available such as gradually twisting synthetic solitons. Continuous nanopatterned layers (i.e., antidot lattices) may be implemented, allowing for discrete electrical address of individual layers. The technological potential of such systems is high, with full compatibility with existing industrial-scale chip fabrication and a host of inviting use cases including sensing, communications and neuromorphic computing. Recent work has shown that expanding the range of accessible microstates and distinct magnonic behaviours in an artificial spin system can greatly enhance neuromorphic computing

capability[45,71], a trend which this 3D metamaterial architecture has high promise to continue.

## Methods

### Micromagnetic simulations

Simulations were performed using MuMax3. Material parameters for NiFe used are; saturation magnetisation, $M_{sat} = 800$ kA/m, exchange stiffness, $A_{ex} = 13$ pJ and damping, $\alpha = 0.001$. All simulations are discretized with lateral dimensions, $c_{x,y} = 4.198$ nm and normal direction, $c_z = 10$ nm and periodic boundary conditions applied to generate lattice from unit cell.

MFM simulations: MFM images are simulated with MuMax3 built-in dipole image function.

Spin-wave/magnon spectra simulations: A broadband field excitation sinc pulse function is applied along $\hat{z}$-direction with cutoff frequency $= 15$ GHz, amplitude $= 1$ mT. Simulation is run for 26 ns saving magnetisation every 33 ps. Static relaxed magnetisation at $t = 0$ is subtracted from all subsequent files to retain only dynamic components, which are then subject to an FFT along the time axis to generate frequency spectra. Power spectra across the field range are collated and plotted as a colour contour plot with resolution; $\Delta f = 18$ MHz and $\Delta \mu_0 H = 1$ mT. For mode profile maps we excite using a sinusoidal function at the resonant frequency of each mode separately with $\alpha = 0.005$. Spatial power maps are generated by integrating over a range determined by the full-width half maximum of peak fits and plotting each cell as a pixel whose colour corresponds to its power. Each colour plot is normalised to the cell with highest power.

Chiral symmetry breaking energy calculations: underlayer is set in a negative $\hat{x}$ saturated state and the top layer is set with either all CW or ACW vortex states. Core polarity was found to have negligible difference in energy and can be considered degenerate. Positive and negative field sweeps are performed separately and then stitched together. Each simulation is performed for different relative offsets between the layers in the $\hat{x}$ and $\hat{y}$ directions. The total energy of the system is calculated in MuMax3 and then combined into the colour contour plots in Fig. 5.

### Semi-analytical model

Calculations for the frequency splitting from a single trilayered 3D nanoisland to the current geometry were performed with the semi-analytical approach 'Gænice'[51]. The method utilizes the Bloch theorem under a tight-binding approximation to compute the Hamiltonian of the system and obtain its eigenfrequencies. In this case, only the static dipole contribution is significant to estimate the FMR frequency when the magnon wavevector $k = 0$. In addition to dipole, we include contributions due to the external magnetic field and demagnetization field. The demagnetization factors for the individual layers in the 3D nanoisland were estimated with Mumax3 and a fitting procedure of the Kittel equation for the bulk modes, detailed elsewhere[72]. The detailed implementation of 'Gænice' is described in ref. 51.

### Experimental methods

**Nanofabrication.** Samples were fabricated via electron-beam lithography liftoff method on a Raith eLine system with PMMA resist. $Ni_{81}Fe_{19}$ (permalloy) and Al were thermally evaporated in alternating layers through the patterned PMMA and capped with 5 nm Al. On-chip CPWs were fabricated for the BLS sample via a similar electron-beam lithography PMMA liftoff process with 300 nm Au deposited and a 3 nm Cr adhesion layer between the Au and the $SiO_2$ substrate.

### Ferromagnetic resonance measurements

FMR spectra were measured using a NanOsc Instruments CryoFMR in a Quantum Design Physical Properties Measurement System.

Broadband differential FMR measurements were carried out on large area samples (~2 × 6 mm²) mounted flip-chip style on a CPW. Nanoislands on the FMR sample have dimensions 550 × 140 × 90 nm and layer thicknesses of 30 nm NiFe/35 nm Al/20 nm NiFe/5 nm Al.

The waveguide is electrically connected to a microwave source, coupling RF magnetic field to the sample and exciting magnon modes. The output from the waveguide was rectified using an RF-diode detector and the rectified voltage was measured via lock-in amplifier. Measurements were done in a fixed in-plane field while the RF frequency was swept in 5–10 MHz steps. The DC field is modulated at 490 Hz with a 0.48 mT RMS field provided by Helmholtz coils and the 490 Hz AC-field perturbation used for the lock-in modulation. The experimental spectra show the derivative output of the microwave signal as a function of field and frequency.

The normalised differential spectra show spin-wave/magnon power as $\frac{\delta P}{\delta \mathbf{H}}$, displayed as false-colour images with symmetric log colour scale and light and dark regions corresponding to strong positive and negative amplitude, respectively. The dark and light bands of the differential peaks can be used to determine the relative magnetisation direction of the magnetic texture giving rise to that mode, from low to high frequency, dark-to-light bands mean the magnetisation texture is oriented in the positive field direction and vice-versa.

### Brillouin light scattering measurements

BLS samples were fabricated on a separate chip, with 8 × 100 μm arrays of nanoislands of dimension 550 × 170 × 90 nm. A micro-patterned on-chip GSG coplanar waveguide of 200 nm thick Au was patterned over the arrays such that the arrays lie between the ground and signal lines to give a $\hat{z}$ RF field. CPW has a signal line of 20 microns width. Micro-focused BLS spectroscopy experiments were conducted in back-scattering geometry (laser incidence normal to the sample plane) using a continuous wavelength single-mode 532 nm laser (Spectra Physics Excelsior). A high-numerical-aperture (NA = 0.75) objective lens with a magnification of 100× collimates the scattered and reflected light. The optical resolution of the used BLS system is less than ≈500 nm. A light source was used to illuminate the sample for monitoring and controlling the measurement position. The inelastically scattered light is analysed using a high-contrast multi-pass tandem Fabry Pérot interferometer (JRS Optical Instruments TFP-2HC) with a contrast of at least $10^{15}$. For all measurements, we plot the Stokes peak of the spectra. All BLS measurements were conducted at room temperature. Spin dynamics detected by BLS were excited by a BNC845 microwave source at nominal powers of +22 dBm.

### Magnetic force microscope measurements

Magnetic force micrographs were produced on a Dimension 3100 using commercially available normal-moment MFM tips. MFM is all performed on the FMR sample.

### Magneto-optical Kerr effect measurements

MOKE measurements were performed on a Durham Magneto-Optics NanoMOKE system. The laser spot is ~20 μm in diameter. The longitudinal Kerr signal was normalised and a linear background contribution arising from the paramagnetic response of the Si substrate and aluminium spacer and capping layers was subtracted from the MOKE signal. The background was determined by taking the signal gradient in the saturated region of the magnetisation loop. The applied field is a quasistatic sinusoidal field cycling at 11 Hz and the measured Kerr signal is averaged over 300 field loops to improve signal-to-noise.

## Data availability
The datasets generated during and/or analysed during the current study are available from the corresponding author on reasonable request.

## Code availability
The code used in this study is available from the corresponding author on reasonable request.

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

## Acknowledgements

This work was supported by the Royal Academy of Engineering Research Fellowships, awarded to J.C.G. T.D. is supported by International Research Fellow of Japan Society for the Promotion of Science (Post-doctoral Fellowships for Research in Japan) JSPS KAKENHI Grant No. 21F20790. J.C.G., W.B. and H.K. were supported by EPSRC grant EP/X015661/1. K.D.S. was supported by The Eric and Wendy Schmidt Fellowship Program and the Engineering and Physical Sciences Research Council (Grant No. EP/W524335/1). Research at the University of Delaware (M.B.J., M.T.K. and R.S.) was supported by the US Department of Energy, Office of Basic Energy Sciences, Division of Materials Sciences and Engineering under Award DE-SC0020308. The authors acknowledge the use of facilities and instrumentation supported by NSF through the University of Delaware Materials Research Science and Engineering Center, DMR-2011824. J.C.G. and M.B.J. were supported by EPSRC ECR International Collaboration Grant 'Three-Dimensional Multilayer Nanomagnetic Arrays for Neuromorphic Low-Energy Magnonic Processing' EP/Y003276/1. H.H. was supported by the EPSRC DTP award EP/T51780X/1. G.A. and E.I. acknowledge support from the National Science Foundation under Grant No. 2205796. A.V. was supported by EPSRC IAA funding. Simulations were performed on the Imperial College London Research Computing Service[73]. The authors would like to thank David Mack for excellent laboratory management and Steve Cussell for excellent technical workshop services.

## Author contributions

J.C.G. and T.D. conceived the work. J.C.G., T.D. and K.S. designed the samples. J.C.G. fabricated the samples. J.C.G. performed ferromagnetic resonance experiments. J.C.G. and T.D. drafted the manuscript, with further contributions from all authors in editing and revision stages. T.D. drafted the descriptions of the micromagnetic simulations. T.D. performed micromagnetic simulations of spin-wave/magnon dynamics, microstate energies and switching pathways, and chiral symmetry breaking energetics. K.S. performed key early micromagnetic simulations of microstate stability vs. layer thicknesses and microstate dimensions. J.C.G. and H.H. performed MFM experiments. A.V. performed magneto-optical Kerr effect experiments. R.S., M.K., J.C.G. and B.J. performed Brillouin light scattering experiments. A.V. wrote code for analysis of ferromagnetic resonance data. G.A., V.M. and E.I. performed semi-analytical mode frequency calculations using Gænice and determined demag factors for the nanoislands by micromagnetic simulations. K.S. produced CGI renders. H.K. and B.J. provided analysis and direction on analysis of the dynamic coupling phenomena. R.O. analysed the lower frequency of optical modes relative to acoustic modes when nanoislands are vertically stacked rather than head-to-head in-plane. H.K., W.B., E.I., T.K., B.J. and J.C.G. provided data analysis, secured funding and provided feedback and discussion on directing the research project.

## Competing interests

The authors declare no competing interests.
