## [Peer Review File · Nature Communications]

Ultrastrong Magnon-Magnon Coupling and Chiral Spin-Texture Control in a Dipolar 3D Multilayered Artificial Spin-Vortex IceREVIEWER COMMENTS

Reviewer #1 (Remarks to the Author):

The authors present results from magnetometry, ferromagnetic resonance and inelastic light scattering for a type of structure constructed from patterned magnetic elements. The geometry is defined by two square artificial ice arrays positioned above one another with an offset along one lateral dimension. The thickness of the elements is different for each array thereby allowing for unambiguous identification of resonances associated with each layer and also facilitating a degree of control over magnetic configurations using applied external fields.

The elements are 30 nm thick, which is sufficient to separate high frequency exchange modes from the lower frequency resonances and localised excitations. Spacing between arrays is varied and evidence for mode hybridisation is presented. A demonstration of the potential for control of magnonic response is provided with an interesting ability to selectively control vortex formation in one layer with the application of a suitable magnetic field.

Whereas the concept of using multilayers for magnonics has been around for some years and explored experimentally and theoretically in a number of papers, the present paper presents a fundamentally new platform that offers far greater flexibility and possibilities for reconfigurability than previously reported. The main results of the paper provide proof of concept through examples of configuration control and in this sense are largely oriented towards enabling the system as a new platform for magnonic device design. Nevertheless, the examples also speak to the potential of designing new experiments that could also provide insight into the physics of magnetic nanoparticles and new studies of systems sufficiently complex to support emergence of unique collective phenomena. Because of this breadth of impact across technology and materials physics, and the increasing interest from several different perspectives in Artificial Spin Ice and related systems, I believe that the paper is suitable for publication in Nature Communications.

The paper is clear, well organised and the figures appropriate and necessary for clarification and illustration of main points. The supporting material is brief and generally helpful especially for understanding the chiral symmetry breaking. I suggest publication and include below questions the authors may wish to consider.

The magnetisation loops shown in Fig. 2(a) show a linear decrease as the field is reduced from high to zero field remanence. Perhaps the authors could comment on this apparent decrease.

Whereas edge modes are identified and briefly discussed, it would be interesting to know more details about these and the types of all modes responsible for the multiple resonance lines observed in the spectra. Micromagnetic generated mode profiles would be helpful.

Related to this, can the authors explain the nature of the fine structure shown in the profiles of Fig. 3d?

The explanation of why optic modes have frequencies lower than acoustic modes is unclear. Naively one might expect that the sign of the splitting would depend on sign of the field energy which in itself depends on the relative orientation of the adjacent magnetic elements (so that in one case out of phase components should lower the mode energy and in the other case raise mode energy).

Finally a comment. The authors present some brief considerations in the supplemental material concerning the linewidths which are used to calculate a magnonic quality factor. This hints at the

potential to design experiments which could in principle be used to provide insight into the surface magnetism of magnetic particles if one could, for example, establish analogous relational comparisons with edge modes and other modes. Likewise temperature and field dependencies might be interesting to study for linewidths. Any related observations concerning the factors contributing to linewidths the authors may have noted during their sample characterisations and optimisations would be interesting to include.

Reviewer #2 (Remarks to the Author):

Dear Editor,

The paper by Dion and coworkers was a difficult read and - in the end - not as rewarding as the title was promising. The difficulty in reading is mainly caused by the fact, that the paper contains 3 different key-messages (three sections) and is constantly exaggerating the observed facts with rather blown-up language. The lack of satisfaction after the read is because the authors promise two fancy sounding discoveries in the title but left me after reading with the feeling, that parts are interpreted incorrectly and/or are described with the wrong words.

I cannot recommend the manuscript for publication in its current state. I am also not convinced that the content is novel/original enough for Nature Communications.

For example: there are two important papers on controlling the circulation of magnetic vortices: The first is a PRB by Dumas and coworkers (Phys. Rev. B 83, 060415 in 2022), where the vortex circulation in disk with a flat side could be controlled by the in-plane direction of the sweeping magnetic field. In principle, this is a very similar mechanism as shown by Dion and it works even in a single-layered nano magnet - so it doesn't require the third dimension. Furthermore: Dion et al claim that they "exploit its dipolar field to selectively control the chirality of magnetic vortex states". However, they never controlled the chirality, because they do not control the polarity of the vortex core. This was for example shown in the second paper by Jain and coworkers in Nature Communications 3, 1330 (2012), where polarity control was demonstrated based on selectively driving coupled vortex gyrosopic modes. Particularly interesting from the point of vortex circulation control is also: Vortex circulation control in large arrays of asymmetric magnetic rings, F. Giesen, J. Podbielski, B. Botters, and D. Grundler, Phys. Rev. B 75, 184428 in (2007).

The authors promise "chiral symmetry breaking". However, their system never had a chiral symmetry. Their vortex is on top of a uniformly magnetized layer introducing a uniaxial bias-field. Hence, no chiral symmetry to begin with.

But most disturbing was the claim of "ultrastrong magnon-magnon coupling" and the calculation of an effective "cooperativity". The authors start the discussion by calculating a coupling strength and mentioning, that they "do not invoke the picture used in some light-matter coupling systems where standard approximations break down and unconserved virtual excitations become active." However, in the next paragraph they write "another metric that can be used to evaluate magnon-magnon coupling strength is the magnon-magnon cooperativity".

The concept of cooperativity is based on the oscillating flow of energy between two well distinguishable physical systems (often seen as anti-crossing of resonance modes in sweeps of a control parameter). This does not make sense within one particular system. Dion et al. analyze a

layered magnetic structure. Both layers are strongly coupled by static and dynamic dipolar magnetic fields. Fine. Agreed. One system with one set of orthogonal eigenmodes. No magnon-magnon cooperativity. I understand that the authors refer to reference [56]. After carefully reading this reference [56] I come to the conclusion that [56] is of questionable quality and doesn't justify the combination of words "magnon-magnon cooperativity". There does exist a phenomenon called magnon-magnon hybridization. This, however, would require the demonstration of a level-crossing or anti-crossing, respectively.

In the phase-maps and the amplitude maps of the mode, which the authors label as acoustic mode, one can see a π phase jump along the short axis of the nano magnet. This is not at all seen in the so-called "optical mode". This phase over the short axis of the nano magnets explain the much higher frequency.

Moreover, the amplitude maxima from top and bottom layer are located at opposite edges. This sort of repulsion of the mode anti-nodes is also seen in the structure of the intensity profile along the long axis of the nano magnet. For example: So-called acoustic mode, lower right structure: Except for the wiggles at the edges, the upper magnet has 3 intensity maxima and in contrast, the lower magnet has always a maximum in between - in total 4. Hence, the names optical and acoustical are most likely attributed to wrong modes.

Regarding the modes with a negative frequency vs field gradient: Similar effects have been observed in arrays of rings. The authors are advised to check the extensive work by Dirk Grundler and coworkers. Also the detection of edge modes has been seen by the same group in electrical measurements. In fully 2D systems. The first paper would be Phys. Rev. Lett. 96, 167207 (2006) and then many thereafter.

In general, the paper lacks quantitative analysis. For example, the authors attribute the static dipolar magnetic fields from the hard layer to blueshift the magnon frequencies in the upper layer. This blueshift could be translated into a magnetic field and this could be compared with the magnetic field extracted from the micro magnetic simulation (but within the other magnets layer and not within the spacer layer).

In Figure 1 and Figure 4 some labels in the caption are just copy paste and do not make sense.

I do not understand the need of sentences like this: "The 3D metamaterial architecture can be considered as n strongly-interacting artificial spin systems separated in z , with $n = 2$ here."

Reviewer #3 (Remarks to the Author):

This is an experimental investigation, supported by micromagnetic simulations, into spin waves in an artificial spin ice structure. The structure is composed of multilayered nanoelements, which consist of two permalloy discs of elliptical shape and varying thicknesses. Moreover, the top permalloy layer is shifted by 50 nm in relation to the central position, and the bottom layer. With this expansion, a significant increase in the number of possible microstates is achieved compared to the standard artificial spin ice system. Furthermore, the potential to control various states using an external magnetic field is shown. This system demonstrates a rich spin-wave spectrum that is strongly dependent on the magnetization configuration, including magneto-toroidal states. The authors assert that ultra-strong interlayer magnon-magnon coupling is evident as a result of dipolar

interactions. This study is both timely and intriguing, expanding the research on artificial spin ice systems and the application of magnonics. However, prior to the decision on its suitability for publication, improvement is necessary.

1. The magnon-magnon coupling is described by the static dipolar interactions between the elements, particularly the Py discs. However, it remains unclear whether and to which extent dynamic dipolar couplings are also involved. It should be noted that photonics, which authors refer to when discussing ultra-strong coupling, involves dynamic coupling between two modes without static effects. In this manuscript, the splitting of acoustic and optic modes seems to be attributed to the mutual modification of the static internal magnetic field of the discs. How reasonable is it to compare it to photonics and name it ultra-strong coupling?
2. In Fig. 1a, the structure shall be shown with the relative shift of the top layers to be consistent with the structure under investigation.
3. Why the thicker Py layer is called "hard" and the thinner "soft"? Seems to be that the thinner has a larger shape anisotropy.
4. There is the statement "The fabrication parameters are tuned to give the strongest inter-layer dipolar coupling possible while keeping all states stable in zero applied magnetic field ($H_{ext} = 0$)."
- What does it mean strongest coupling and how it was determined?
5. From the caption of Fig. 2f it is not clear meaning of the color map and dots.
6. From fig. 6 a and b it seems that the agreement between measurements and simulation results is far from being perfect. What can be a reason? In BLS spectra, fig 2f, the high-frequency band as a function of H in the antiparallel state seems to be nonmonotnous, while in the parallel state two intensive bands are visible, which does not seem to be present in simulation results, why?
7. Page 7, probably in the last two paragraphs references to figures shall be corrected. Also on p. 10, second paragraph.
8. The modes identified on p. 8: "An extremely rich spectra is observed, given by a combination of magnetic vortex states (χ -shaped modes, 2-6 GHz). edge-curvature states (dominant edge-curvature mode overlaps with the 2-3 GHz vortex modes, and 3 higher-frequency edge-curvature modes 3-5 GHz)" are very difficult to find in the figure.
9. Which magnetisation components were used to calculate the power spectra?
10. The caption of Fig. 7 in Supl. Mat. Needs to be extended to describe in detail all presented plots.
11. Fig. 11 and 12: The color scale for the demagnetizing field values is missing.

REVIEWER COMMENTS

Reviewer #1 (Remarks to the Author):

The authors present results from magnetometry, ferromagnetic resonance and inelastic light scattering for a type of structure constructed from patterned magnetic elements. The geometry is defined by two square artificial ice arrays positioned above one another with an offset along one lateral dimension. The thickness of the elements is different for each array thereby allowing for unambiguous identification of resonances associated with each layer and also facilitating a degree of control over magnetic configurations using applied external fields.

The elements are 30 nm thick, which is sufficient to separate high frequency exchange modes from the lower frequency resonances and localised excitations. Spacing between arrays is varied and evidence for mode hybridisation is presented. A demonstration of the potential for control of magnonic response is provided with an interesting ability to selectively control vortex formation in one layer with the application of a suitable magnetic field.

Whereas the concept of using multilayers for magnonics has been around for some years and explored experimentally and theoretically in a number of papers, the present paper presents a fundamentally new platform that offers far greater flexibility and possibilities for reconfigurability than previously reported. The main results of the paper provide proof of concept through examples of configuration control and in this sense are largely oriented towards enabling the system as a new platform for magnonic device design. Nevertheless, the examples also speak to the potential of designing new experiments that could also provide insight into the physics of magnetic nanoparticles and new studies of systems sufficiently complex to support emergence of unique collective phenomena. Because of this breadth of impact across technology and materials physics, and the increasing interest from several different perspectives in Artificial Spin Ice and related systems, I believe that the paper is suitable for publication in Nature Communications.

We thank the referee for their positive assessment of our work, and for their recommendation of publication in Nature Communications.

The paper is clear, well organised and the figures appropriate and necessary for clarification and illustration of main points. The supporting material is brief and generally helpful especially for understanding the chiral symmetry breaking. I suggest publication and include below questions the authors may wish to consider.

We again thank the referee for their comments on the clarity and presentation of our work. We find all the referee's suggestions positive and constructive, and are happy to make all the suggested changes.

1A The magnetisation loops shown in Fig. 2(a) show a linear decrease as the field is reduced from high to zero field remanence. Perhaps the authors could comment on this apparent decrease.

The linear background in the MOKE data is due to paramagnetic contributions to the MOKE signal from both the aluminium spacer layer & anti-oxidation capping layer, and the Si substrate (the SiO₂ layer on top of the Si is relatively transparent at the 633 nm wavelength of the laser used for the MOKE experiments). The magnetisation of the NiFe nanostructures is not changing linearly with the field above ~41 mT where the magnets are in a saturated state.

This linear paramagnetic background contribution is a commonly observed artefact in MOKE experiments and is typically removed by subtracting a linear term, with the linear term obtained from a fit to the saturated regions of the curve.

We neglected to perform this subtraction before submission, and we thank the referee for bringing this point to our attention. Below we show the corrected data which is now included in fig 2. The linear correction works well, with the corrected MOKE signal now more accurately reflecting the magnetic response of the NiFe nanostructures.

The following comment has been added to the MOKE methods section:

“The longitudinal Kerr signal was normalised and a linear background contribution arising from the paramagnetic response of the Si substrate and aluminium spacer and capping layers was subtracted from the MOKE signal. The background was determined by taking the signal gradient in the saturated region of the magnetisation loop.”

1B Whereas edge modes are identified and briefly discussed, it would be interesting to know more details about these and the types of all modes responsible for the multiple resonance lines observed in the spectra. Micromagnetic generated mode profiles would be helpful.

We thank the referee for the suggestion, and agree that adding more detail on the mode profiles would improve the paper. We have included these as a new panel f) in figure 2 and accompanying discussion in the text. The new panel shows the spatial power profiles of the various edge (labelled E) and nanoisland-centre localised modes (labelled B for nanoisland 'bulk') for the different magnetisation states. Five modes (three edge, two bulk) and corresponding profiles are shown for the parallel macrospin states, with three modes (one edge, two bulk) for the antiparallel state. The antiparallel state has fewer edge modes as the

attractive interactions between upper and lower magnetic nanoisland layers straighten out the edge-curvature of the magnetisation and suppress the edge-modes, whereas the repulsive interactions in the parallel macrospin state push the magnetisation at nanoisland ends into a curved state.

A section of the updated figure 2 is shown below.

Related to this, can the authors explain the nature of the fine structure shown in the profiles of Fig. 3d?

The referee raises a good point on the fine structure of the spatial mode power and phase maps of the acoustic and optical modes. Previously we used a relatively low damping parameter of 0.001 to ensure all possibly active modes were excited. However, the lower damping can create unrealistically complex phase profiles due to interaction with modes that would be damped/not excited in the real system. We have re-run the simulations with a more physically realistic damping parameter of 0.005 and we now see smoother and cleaner mode profiles.

We also previously used a broadband excitation across a wide range of frequencies by using a simulated RF field modelled by a sinc pulse in time. This allows us to excite all frequencies with a single simulation, computationally efficient, but the multiple concurrently excited modes can cause some interference between modes at different frequencies and result in complex spatial power and phase profiles.

We have now replaced the sinc pulse function with two separate simulations using sine wave RF fields, one at 6.267 GHz (optical), one at 10.512 GHz (acoustic). The sine-excitation case gives cleaner mode power and phase maps than the broadband sinc pulse case.

We plot the updated data below, alongside time traces which show the evolution of magnetisation for the top and bottom magnetic layers, integrated over a whole nanoisland. The time traces very clearly show the out-of-phase and in-phase relations for the optical and acoustic modes respectively.

1C The explanation of why optic modes have frequencies lower than acoustic modes is unclear. Naively one might expect that the sign of the splitting would depend on sign of the field energy which in itself depends on the relative orientation of the adjacent magnetic elements (so that in one case out of phase components should lower the mode energy and in the other case raise mode energy).

The referee raises a good point & we're happy to provide further clarification here. From our understanding there are two related questions here: why does the optical mode have a lower frequency than the acoustic mode, and why is no optical mode resolved in the parallel state?

The key point to understanding whether the acoustic or optical mode should have a higher relative energy and frequency is the geometry/angle between the nanoislands - eg. are the elements coupled tip-to-tip (such as adjacent islands in a normal 2D ASI, or spins in an antiferromagnetic chain) or side-by-side (as in our vertically stacked nanoislands). Below is a schematic illustrating these two geometries:

Because of our vertical 'side-by-side' arrangement, the strongest dipole coupling arises from the inter-layer component. Energetically, dipolar coupling leads to lower-energy modes in the out-of-phase configuration. This is different than 'tip-to-tip' configurations more commonly

used in magnonics where the in-phase configuration has a lower energy. We have studied such a 'tip-to-tip' hybridisation previously in a single-layer 2D ASI, and the optical mode in that case is higher frequency than the acoustic mode, as expected. Relevant papers are *Dion, T., et al. "Observation and control of collective spin-wave mode hybridization in chevron arrays and in square, staircase, and brickwork artificial spin ices." Physical Review Research 4.1 (2022): 013107.* and *Gartside, Jack C., et al. "Reconfigurable magnonic mode-hybridisation and spectral control in a bicomponent artificial spin ice." Nature Communications 12.1 (2021): 2488.* A similar effect occurs in the field of plasmonics in coupled oscillations between two nearby nanoislands, where the coupling is also dipolar in origin. A thorough investigation of how the geometric arrangement of the two coupled islands will invert the relative energies and frequencies of the hybridised acoustic and optical modes can be found in "Davis, T. J., D. E. Gómez, and K. C. Vernon. "Simple model for the hybridization of surface plasmon resonances in metallic nanoparticles." *Nano letters* 10, no. 7 (2010): 2618-2625.", particularly in figure 2.

The remaining question is why don't we observe an optical mode in the parallel magnetised macrospin case? To drive an out-of-phase oscillation of two moments, one needs to exert an unequal/opposite torque on each moment. In the case of a spatially uniform driving field, as is the case here, this can only be achieved by applying the uniform RF field along a spatial axis where the magnetic symmetry is broken. This is the case when preparing an antiparallel state, & applying the RF field along the magnetisation axis, eg the RF field must be applied along the same axis as the DC H field, a geometry often termed parallel or parametric pumping. Here, the two magnetic layers in an antiparallel state experience opposite torques and the out of phase optical mode may be excited.

For a spatially symmetric parallel magnetic state, there is no spatial axis where a uniform RF field can exert unequal torques on the two magnetic layers, hence only the in-phase acoustic mode may be excited in the parallel magnetic state.

1D Finally a comment. The authors present some brief considerations in the supplemental material concerning the linewidths which are used to calculate a magnonic quality factor. This hints at the potential to design experiments which could in principle be used to provide insight into the surface magnetism of magnetic particles if one could, for example, establish analogous relational comparisons with edge modes and other modes. Likewise temperature and field dependencies might be interesting to study for linewidths. Any related observations concerning the factors contributing to linewidths the authors may have noted during their sample characterisations and optimisations would be interesting to include.

This is a good suggestion, and we agree with the referee that this is a study which would provide further insight into the behaviour of the system. We would be interested in conducting such a study as a separate future work.

We believe that the results of such a study may get lost if added to the current manuscript, there is not space in the main text so the new results would end up as supplementary information. Conducting these further experiments at a range of temperature and field states will take a substantial amount of experimental time and post-experiment analysis, so we believe this study is best suited as an independent separate future work. We thank the referee again for the suggestion of research direction.

Overall we thank the reviewer for the time taken to put together their review, for their constructive suggestions and for the opportunity to improve our manuscript.

Reviewer #2 (Remarks to the Author):

Dear Editor,

The paper by Dion and coworkers was a difficult read and - in the end - not as rewarding as the title was promising. The difficulty in reading is mainly caused by the fact, that the paper contains 3 different key-messages (three sections) and is constantly exaggerating the observed facts with rather blown-up language. The lack of satisfaction after the read is because the authors promise two fancy sounding discoveries in the title but left me after reading with the feeling, that parts are interpreted incorrectly and/or are described with the wrong words.

I cannot recommend the manuscript for publication in its current state. I am also not convinced that the content is novel/original enough for Nature Communications.

We thank the reviewer for taking the time to review our work, and for their thorough assessment and suggestions. We have now answered all the points raised, and our manuscript is much improved as a result. We thank the reviewer for this opportunity to improve the quality of our work.

2A For example: there are two important papers on controlling the circulation of magnetic vortices: The first is a PRB by Dumas and coworkers (Phys. Rev. B 83, 060415 in 2022), where the vortex circulation in disk with a flat side could be controlled by the in-plane direction of the sweeping magnetic field. In principle, this is a very similar mechanism as shown by Dion and it works even in a single-layered nano magnet - so it doesn't require the third dimension.

The referee correctly mentions that methodologies for controlling the vortex circulation by patterning asymmetric nanoelements have been previously demonstrated. We fully agree that these should be referenced and have expanded our discussion on the chirality control mechanism to give proper credit to the various solutions for controlling vortex circulation and chirality. We thank the referee for the suggestion.

It is not our claim that controlling the vortex chirality is a new result - or that the 3rd dimension is required for chirality control. Our focus in this work is that introducing the third dimension to nanomagnetic arrays can give an expanded range of microstates and magnonic behaviours. One might think that by increasing the range of microstates from 2^N in a conventional ASI array up to 16^N , one would lose some ability to selectively prepare these different states.

Our intention with the chirality control mechanism was to show that alongside the expanded range of states given by the multilayer architecture, we also gain new means to control these states: using the reconfigurable local dipolar field from an adjacent macrospin layer to break energetic symmetry between different CW/ACW vortex states and hence access the lower energy state. Using this method we can program desired CW/ACW vortex states, or

deactivate the selectivity mechanism to program a stochastic mixed-chirality state if desired. We have added the below clarifying statement & references to the manuscript:

“A number of methods exist to control the state of magnetic vortices in nanostructures, typically involving breaking the symmetry of the nanostructure itself via fabricating an asymmetrically-shaped structure. Examples include disks with a sliced-off edge to give a 'D' shape (schneider2001magnetic, kimura2007vortex, dumas2011chirality}, variable-width crescent-shaped rings (giesen2007vortex), merging two disks together (jain2012chaos) and asymmetric bicomponent wedge rings (shimon2012reversal). This suite of approaches allows powerful control over CW/ACW vortex circulation and in some cases core polarity. The catch is that as the nanostructures themselves are asymmetric, different vortex states have asymmetry introduced into their dynamics and energy. This asymmetry is hard-coded at the fabrication stage and may not be deactivated or reconfigured.

A benefit of the approach described here is that by using the dipolar field of an adjacent 3D nanoisland layer to control vortex states, the nanostructures may remain symmetric and the dipolar field may be reconfigurably programmed on a per-island basis, for instance by using local magnetic writing (gartside2018realization,stenning2023low). The dipolar field control may also be effectively deactivated by preparing the control layer in a vortex state that has far lower stray dipolar field than a macrospin state.”

Furthermore: Dion et al claim that they "exploit its dipolar field to selectively control the chirality of magnetic vortex states". However, they never controlled the chirality, because they do not control the polarity of the vortex core. This was for example shown in a the second paper by Jain and coworkers in Nature Communications 3, 1330 (2012), where polarity control was demonstrated based on selectively driving coupled vortex gyroscopic modes. Particularly interesting from the point of vortex circulation control is also: Vortex circulation control in large arrays of asymmetric magnetic rings, F. Giesen, J. Podbielski, B. Botters, and D. Grundler, Phys. Rev. B 75, 184428 in (2007).

The authors promise "chiral symmetry breaking". However, their system never had a chiral symmetry. Their vortex is on top of a uniformly magnetized layer introducing a uniaxial bias-field. Hence, no chiral symmetry to begin with.

The referee makes a fair point on the choice of language here. The referee is correct that the chirality of a magnetic vortex state is defined as a combination of its in-plane chiral circulation and its out-of-plane core polarity. In this study, due to the multilayered composition of our structures it is challenging to resolve the core polarity from our magnetic force microscope data, and we do not intend to indicate that we have experimentally demonstrated control over the core polarity state.

In response to the referees comment, we have performed additional micromagnetic simulations (mumax3) comparing the energetics of different core polarity states. Below we show a symmetry breaking/energetic degeneracy lifting between vortex states of the same chirality but opposite core polarity. The colour bar shows the energetic difference as a fraction of the combined demagnetisation and exchange energy of the whole multilayer nanoisland.

The system is seen to energetically favour one core polarity over the other, with an energetic degeneracy lifting of 0.3% of the entire multi-layered system energy. This is a smaller %

value than for the CW/ACW degeneracy lifting, which is understandable as the vortex core represents a much smaller volume fraction of the system relative to the in-plane CW/ACW magnetisation regions.

In light of the new simulation results, we demonstrate energetic symmetry breaking between both CW/ACW circulation and up/down core polarity. As such, we believe it is reasonable to state that this system shows chiral symmetry breaking. We have included these new simulation results into figure 4.

As mentioned above, resolving the vortex core polarity experimentally via MFM is challenging in this multilayered system, and this would require substantial additional work beyond the scope of this study. There is a likelihood that the energetic symmetry breaking is driving core polarity selectivity in the same manner as it is driving CW/ACW vortex selectivity, and this is a promising avenue for experimental confirmation in future work.

We have added the new simulations on core polarity to figure 4 and added the following text to the manuscript text in the discussion of figure 4:

“The full chirality of magnetic vortex state may be characterised as combination of the ‘circulation’ (the CW/ACW chirality of the in-plane magnetic texture) and the vortex core polarity (+-z ‘up/down’ component at the centre of the vortex). The bottom two panels of figure 4b) show that in the case of an x-direction inter-layer offset and x-direction H_{ext} , symmetry is broken for +-z ‘up/down’ vortex core polarities. An energy difference up to 0.3% of the total system energy is observed, smaller than the CW/ACW energy difference which is as expected due to the vortex core comprising a much smaller volume fraction of the system relative to the in-plane magnetisation regions.”

“It is challenging to resolve the vortex core polarity via MFM in this multilayered system. From our simulation of the energetic degeneracy lifting between core polarity states, there is a likelihood we are controlling vortex polarity and circulation state but full confirmation of this requires further experimental work.”

In terms of the referee’s comment that there was no chiral symmetry to begin with due to the presence of the collinear macrospin island, we did not intend to suggest that the whole multilayer system as a whole was chirally symmetric. We mean only to state that the nanoisland layer which is prepared in a vortex state is chirally symmetric.

We clarify here that the chiral symmetry breaking which we refer to is the reconfigurable control over whether the system has equal likelihood of assuming CW or ACW vortex circulation (eg. in the case of fig 4 b,e,g,i), or whether this symmetry is broken and the

system is programmed to only assume a specific desired CW or ACW vortex state (eg. in the case of fig 4 a,c,d,f,j,k). We have clarified in the text that we are referring to chiral selectivity over the layer prepared in the vortex state with the following statement:

“By placing one magnetic layer in a macrospin state, we may exploit its dipolar field to selectively control the chiral CW/ACW state of magnetic vortices in the adjacent layer. This chiral selectivity may be activated or deactivated on-demand by programming the state of the macrospin layer, allowing three regimes: forced CW vortices, forced ACW vortices and stochastic mixed chirality.”

2B But most disturbing was the claim of "ultrastrong magnon-magnon coupling" and the calculation of an effective "cooperativity". The authors start the discussion by calculating a coupling strength and mentioning, that they "do not invoke the picture used in some light-matter coupling systems where standard approximations break down and unconserved virtual excitations become active." However, in the next paragraph they write "another metric that can be used to evaluate magnon-magnon coupling strength is the magnon-magnon cooperativity".

The concept of cooperativity is based on the oscillating flow of energy between two well distinguishable physical systems (often seen as anti-crossing of resonance modes in sweeps of a control parameter). This does not make sense within one particular system. Dion et al. analyze a layered magnetic structure. Both layers are strongly coupled by static and dynamic dipolar magnetic fields. Fine. Agreed. One system with one set of orthogonal eigenmodes. No magnon-magnon cooperativity. I understand that the authors refer to reference [56]. After carefully reading this reference [56] I come to the conclusion that [56] is of questionable quality and doesn't justify the combination of words "magnon-magnon cooperativity". There does exist a phenomenon called magnon-magnon hybridization. This, however, would require the demonstration of a level-crossing or anti-crossing, respectively.

The referee raises a good point that cooperativity is typically used to describe oscillating energy transfer between two dissimilar systems. It is indeed correct that the term is invoked generally in situations where different quasi-particles are involved in the energy transfer process, eg. magnon-phonon or magnon-photon coupling. We are happy to amend our manuscript here, and agree with the referee that potential confusion may be avoided by removing it.

In terms of providing additional evidence in support of our claims of ultrastrong magnon-magnon coupling, we are happy to share additional data showing the opening and closing of an anti-crossing gap in response to sweeping a control parameter as the referee mentions. To be clear, the magnon-magnon coupling in our system is between magnons in the upper magnetic layer and magnons in the lower magnetic layer, mediated by dynamic dipolar coupling between the layers.

Control parameters which are typically used to examine anti-crossing gaps in magnonic systems are the external magnetic field magnitude or field angle relative to the system magnetisation. However in our system, the coupling gap is not created via an externally applied field, rather it is created via the local inter-layer dynamic dipolar field of the nanomagnets. The control parameter we sweep to demonstrate the opening of an

anti-crossing gap and show mode hybridisation is the thickness of the non-magnetic spacer layer between the two NiFe layers, essentially the inter-layer separation:

As the referee states, both magnetic nanoisland layers are strongly coupled via dynamic dipolar field. Indeed, the lack of reliance on an externally applied magnetic field is a key attractive feature of our results: the process is local & passive, with the required dipolar field supplied by the multilayered nanoislands themselves.

However, the other side of this is that it is not easy to sweep the more typical control parameters used in magnonic experiments: the angle between moments or the magnitude of this local dynamic dipolar field in experiment. The angle between moments is constrained to a quantised 0 or 180 degrees by the shape anisotropy of the nanomagnets, and the magnitude is constrained by the thickness of the non-magnetic spacer layer and the thickness of the magnetic layers. Hence, our parallel and antiparallel magnetisation states represent a fully closed or fully open gap respectively. To vary the magnitude of the anticrossing gap, the separation between the magnetic layers would need to be varied as we have shown above in simulation - not possible experimentally without fabricating many new samples of different spacer layer thicknesses.

The investigation of spacer layer thicknesses was included in the original manuscript, but just as extracted anti-crossing gap width values which are less visually instructive than the new plots shown above. The original figures 3g) and 3h) showed the gap going from fully closed at a separation of 250 nm, to ~ 7.6 GHz at a separation of 5 nm. For convenience we have copied these two panels below:

We have substantially updated figure 3 to better illustrate our magnon-magnon coupling results, we believe it is much improved as a result. We have included magnetisation vs time

traces for the acoustic and optical modes, showing the time evolution of the averaged magnetisation of a single nanoisland for the top and bottom magnetic layers (panel 3i). The time traces very clearly show the in-phase and out-of-phase oscillation behaviour for the acoustic and optical modes respectively. We show the new figure 3 below:

We have also performed anti-crossing gap simulations at many different spacer layer thicknesses and included this as a supplementary figure to give full detail to interested readers. This is shown below.

If we examine the mode power profiles for the 250 nm case where modes have no anti-crossing and are not hybridised, as expected we see that each mode corresponds to magnons active in a single layer only:

This shows we have no inter-layer magnon coupling or hybridisation at larger spacer thicknesses, and magnon-magnon coupling and hybridisation and anti-crossing gap generation occur at thinner spacer layer thicknesses where the dynamic component of the inter-layer dipolar field becomes sufficiently intense to generate strong magnon-magnon coupling effects.

As further supporting evidence that our higher and lower frequency modes are acoustic and optical in character, below we show two plots of experimental data comparing the magnon spectra when the RF field is oriented perpendicular and parallel to the uniform static externally applied magnetic field H respectively:

The lower frequency optical mode is observed strongly and clearly when the static H field and RF field are parallel, with only a very weak hint of a mode observed when the H and RF fields are perpendicular. This effect is observed in synthetic antiferromagnets, where only the parallel H and RF field case is able to couple to the out-of-phase optical mode. The reason that the optical mode is strongly resolved when the Rf field is parallel to the H field is that to drive an out-of-phase optical mode, an unequal/opposite torque must be exerted on the two coupled moments. The moments are aligned anti-parallel along the applied H field direction, hence an RF field applied along this axis will exert opposite torques on the two coupled moments. When the RF field is applied perpendicular to this axis, an equal torque is applied to both moments as they are both equally orthogonal to the RF field axis. In this case only the in-phase acoustic mode may be efficiently coupled to and excited.

An example of this effect is shown below, taken from Sud, A., et al. "Tunable magnon-magnon coupling in synthetic antiferromagnets." *Physical Review B* 102.10 (2020): 100403. Panels b) and d) show the optical and acoustic modes excited when the B field is parallel to the RF field, and panels c) and e) show only the acoustic mode excited when the B field is perpendicular to the RF field:

The reason we observe a hint of the optical mode in the perpendicular H and RF field case is that our nanoislands are oriented at 45 degrees to the applied H field. As such, there is still some component of the nanoisland magnetisation which satisfies the parallel field geometry condition, allowing a weak coupling of the RF field to the optical mode and the corresponding relative weakly observed optical mode. In the thin-film case of synthetic antiferromagnets, there is no such shape anisotropy and all magnetisation lies along the applied H field axis.

To summarise, we have demonstrated the following evidence for the acoustic/optical character of our hybridised modes and magnon-magnon coupling between layers:

- Opening of an anti-crossing gap by varying the inter-layer spacing.
- Demonstration that at larger (>250 nm) inter-layer spacing thicknesses, modes cross and each mode corresponds to magnons in a single magnetic layer oscillating alone.
- Demonstration that at smaller (<150 nm) inter-layer spacing thicknesses, modes anti-cross and each mode corresponds to both upper & lower magnetic layers oscillating together, eg. hybridisation between magnons in the upper and lower layers.
- Demonstration that the optical mode is strongly excited when the RF field is parallel to the applied DC H field, and not when the RF field is perpendicular to the DC H field.
- Updated simulated phase maps and magnetisation vs. time traces showing in-phase behaviour for the acoustic mode and out-of-phase behaviour for the optical mode (shown further below in response to comment 2C).

We believe that with this set of complementary results, it is reasonable to say that the magnon modes in our upper and lower magnetic nanoisland layers are sufficiently coupled via the dynamic dipolar field to hybridise & generate acoustic and optical modes, with measurements of the coupling strength quantitatively in the ultrastrong magnon-magnon coupling regime.

We thank the referee for the opportunity to perform additional work to clarify and strengthen our argument. We believe the manuscript is much improved by this additional work.

2C In the phase-maps and the amplitude maps of the mode, which the authors label as acoustic mode, one can see a π phase jump along the short axis of the nano magnet. This is not at all seen in the so-called "optical mode". This phase over the short axis of the nano magnets explain the much higher frequency.

Moreover, the amplitude maxima from top and bottom layer are located at opposite edges. This sort of repulsion of the mode anti-nodes is also seen in the structure of the intensity profile along the long axis of the nano magnet. For example: So-called acoustic mode, lower right structure: Except for the wiggles at the edges, the upper magnet has 3 intensity maxima and in contrast, the lower magnet has always a maximum in between - in total 4. Hence, the names optical and acoustical are most likely attributed to wrong modes.

The referee raises a good point on the phase maps of the acoustic and optical modes in fig 3. We have responded to this point in response to reviewer 1 above, for ease of reading we repeat our response here.

The referee raises a good point on the fine structure of the spatial mode power and phase maps of the acoustic and optical modes. Previously we used a relatively low damping parameter of 0.001 to ensure all possibly active modes were excited. However, the lower damping can create unrealistically complex phase profiles due to interaction with modes that would be damped/not excited in the real system. We have re-run the simulations with a more physically realistic damping parameter of 0.005 and we now see smoother and cleaner mode profiles.

We also previously used a broadband excitation across a wide range of frequencies by using a simulated RF field modelled by a sinc pulse in time. This allows us to excite all frequencies with a single simulation, computationally efficient, but the multiple concurrently excited modes can cause some interference between modes at different frequencies and result in complex spatial power and phase profiles.

We have now replaced the sinc pulse function with two separate simulations using sine wave RF fields, one at 6.267 GHz (optical), one at 10.512 GHz (acoustic). The sine-excitation case gives cleaner mode power and phase maps than the broadband sinc pulse case.

We plot the updated data below, alongside time traces which show the evolution of magnetisation for the top and bottom magnetic layers, integrated over a whole nanoisland. The time traces very clearly show the out-of-phase and in-phase relations for the optical and acoustic modes respectively.

2D Regarding the modes with a negative frequency vs field gradient: Similar effects have been observed in arrays of rings. The authors are advised to check the extensive work by Dirk Grundler and coworkers.

We thank the referee for this suggestion, we have cited the nice work by Grundler et al “Giesen, F., Podbielski, J., Botters, B., & Grundler, D. (2007). Vortex circulation control in large arrays of asymmetric magnetic rings. *Physical Review B*, 75(18), 184428.”

Also the detection of edge modes has been seen by the same group in electrical measurements. In fully 2D systems. The first paper would be Phys. Rev. Lett. 96, 167207 (2006) and then many thereafter.

In terms of detecting edge modes, we do not claim to detect edge modes for the first time and the referee is quite right that edge modes have been studied before. The 2006 PRL that the referee mentions is a high quality study. However the 2006 PRL the referee quotes is a study on nanoring systems which by definition do not have the curved C-state or S-state edge magnetisation states at the end of nanoislands which are the edge states we refer to in this work. An example of the nanoisland edge states which we refer to in the manuscript is this work by Sloetjes et al: “Sloetjes, Samuel D., Björgvin Hjörvarsson, and Vassilios Kapaklis. "Texture fluctuations and emergent dynamics in coupled nanomagnets." *Physical Review B* 106, no. 10 (2022): 104405.”

The interesting result which we show here, is that the interlayer coupling between the upper and lower nanoisland layers can be exploited to exaggerate the degree of C/S state edge curvature relative to a conventional single-layer magnetic nanostructure - which results in far stronger edge-modes than are typically observed in a single-layer system. This exaggerated curvature and corresponding strong edge-modes are also reconfigurable. This is a consequence of the multilayer nanoisland architecture and a good example of the additional controls & behaviours enabled by implementing such a system geometry. Below is a reproduction of the figure corresponding to supplementary note 7 where we show this reconfigurable behaviour:

FMR: Parallel Macrospin State
Positive Saturation, Positive field sweep

FMR: Parallel Macrospin State
with Edge-Curvature
Positive Saturation, Positive field sweep

Main FMR mode

Edge modes

The panel on the left shows the trilayer system in a parallel macrospin state with a normal degree of S/C-state edge curvature, typical of a single-layer ASI. While edge modes relating to this end curvature do exist & are seen in simulation, they are relatively weak in signal-to-noise, far weaker than the main FMR mode at 10-12 GHz and challenging to detect experimentally. In our FMR experiment they are below the noise floor.

The panel on the right shows the trilayer system in a parallel macrospin state with an exaggerated degree of edge curvature, accessed by first preparing an antiparallel macrospin state, then reversing one layer back into a parallel state. Here, the edge curvature of the magnetisation ends up in a much more strongly curved metastable state, with a corresponding pronounced edge-mode that has similar amplitude to the main FMR mode and is very easily detected, alongside a higher-order edge mode at higher frequency. The curvature in the top and bottom layers are in opposite directions. Above ~43 mT, the strong edge curvature straightens out to point in the same direction in both layers matching the

magnetisation state shown in the left column above, the pronounced edge mode disappears and the left and right spectra are correspondingly identical at fields above ~43 mT.

Notably the main FMR mode at 10-12 GHz is unaffected by switching the system between the two metastable edge magnetisation states, demonstrating that the switchable edge states in this system serve as an additional degree of freedom beyond the macrospin/vortex state selection.

We are also able to remove the lower frequency edge modes by preparing the system in the anti-parallel state, where the strong dipolar attraction between oppositely-magnetised nanoisland edges in adjacent layers leads to straight edge magnetisations without curvatures. This is seen in the simulated magnetisation maps and absence of the lower-frequency edge modes in figure 2.

To sum up this point, we do not suggest that we are the first to detect edge modes, or that edge mode detection is novel. The novel result is that our system has reconfigurable edge magnetisation states as a consequence of its 3D geometry, and when switching between these distinct edge magnetisation states we observe a strong reconfigurable change in the magnon spectra and mode profiles.

We have added the following comment to the text:

“Detecting edge-modes in nanostructures can be challenging relative to main Kittel modes due to their small volume fraction and sensitivity to nanofabrication imperfections. The strong 3D dipolar coupling in this system increases the volume of the curved edge regions in the parallel macrospin state and hence renders them easily resolvable relative to a single-layer system, with the 2.6 GHz edge mode in this state having equal amplitude to the main Kittel mode at 10.6 GHz.

In the antiparallel state, edge magnetisation no longer curves due to the strong dipolar attraction to the edge of the adjacent nanoisland layer. This is reflected by the absence of lower-frequency edge modes in the antiparallel state, seen in figures 2 b), e) and f).”

2E In general, the paper lacks quantitative analysis. For example, the authors attribute the static dipolar magnetic fields from the hard layer to blueshift the magnon frequencies in the upper layer. This blueshift could be translated into a magnetic field and this could be compared with the magnetic field extracted from the micro magnetic simulation (but within the other magnets layer and not within the spacer layer).

This is a good suggestion, we thank the reviewer for mentioning it. We have now included a comparison of the dipolar field distribution (generated from simulations) and evaluation of the applied magnetic field required to generate frequency shifts as the reviewer suggests. We show the new data below.

We have simulated two cases: a 30 nm thick NiFe nanoisland (left four columns) and a 20 nm thick NiFe nanoisland (right four columns). We then plot the x, y, z and combined $\sqrt{x^2+y^2+z^2}$ components of the stray dipolar field (columns are labelled at the bottom of each column), with each row showing the dipolar field for a 10 nm z slice. The substrate is at the bottom of the column, in the 30 nm island case (left four columns) the upper magnetic layer will be situated in the top two rows, in the 20 nm island case (right four columns) the

upper magnetic layer will be situated in the top three rows. The position the upper magnetic layer would occupy is shown by curly brackets. To clarify, we are only simulating a single magnetic layer at a time to extract the dipolar field contribution from each layer separately.

From the total field column, we see there is a peak field at the position of the adjacent magnetic layer of 60–80 mT in the 30 nm island case, and a peak field of 35–45 mT in the 20 nm island case. These fields are highly spatially nonuniform, concentrated above the nanoisland ends as expected for a dipolar field projected from the poles of a macrospin nanoisland. As the mode is located at the centre of the bar the field here will contribute the most to the shift in frequency, which is lower than at the poles. The field component along the long axis of the nanoislands around the nanoisland centre is $\sim 15\text{--}17$ mT in the case of the 20 nm island and $18\text{--}21$ mT in the case of the 30 nm island, with this variation due to the spatially-nonuniform nature of the dipolar field.

From Fig 3a) we can use the experimental FMR data to look at the amount of spatially-uniform externally-applied magnetic field required to generate a shift of 1 GHz (the difference in mode resonance between the parallel and antiparallel macrospin states).

From a resonance of 11 GHz at 20 mT and a resonance of 12 GHz at 53 mT, we can model the mode frequency/field gradient in this region as 1 GHz / 33 mT using a linear first-order approximation.

The experimentally-measured blueshift in mode-frequency between the parallel and antiparallel states is 1 GHz. The shift in dipolar field between these states is 15-17 + 18-21 mT = 33-38 mT. This is a good match for the experimentally estimated value of 33 mT for a 1 GHz frequency shift.

We have added the new data on the stray dipolar field to the supplementary information alongside accompanying discussion in the supplementary information. The following commentary has been added to the main text:

“This frequency shift arises from the static component of the stray dipolar field, and we can compare the magnitude of frequency shift to the magnitude of static dipolar field. Micromagnetic simulations of the spatially-distributed dipolar field magnitude (shown in supplementary figure 7) give a difference in static dipolar field between parallel and antiparallel macrospin states of 33-38 mT. Comparing this with the field-swept FMR data in fig. 2 b) and using a first-order approximation of a linear field/frequency gradient of the Kittel mode, we see that 33 mT of applied H_{ext} field is required to generate a mode shift of 1 GHz - a good correspondence with the simulated dipolar field magnitude”.

2F In Figure 1 and Figure 4 some labels in the caption are just copy paste and do not make sense.

We believe the referee is referring to caption labels such as this example from figure 1: “(d-g) 3D coupled nanoisland magnetisation states. Micromagnetic simulations of top (d) and bottom (e) layer nanoislands are shown alongside simulated (f) and experimental (g) MFM

contrast taken at 50 nm above the 3D nanoarray. Experimental MFM of a state containing both parallel and antiparallel macrospin states is shown in the bottom left, demonstrating the large difference in stray dipolar field between parallel (strong contrast, high dipolar field) and antiparallel (weak contrast, low dipolar field) macrospin states”

Here, to keep the caption concise we introduce related panels together, but separately define and reference all panels in the text, eg. the separate (d), (e), (f) and (g) descriptions in the text of the caption label.

We believe this is a reasonable way to display caption information, however if the journal editor would like us to amend this at the copy editing stage we are happy to do so.

2G I do not understand the need of sentences like this: "The 3D metamaterial architecture can be considered as n strongly-interacting artificial spin systems separated in z, with n = 2 here."

We thank the reviewer for bringing to light the potentially confusing language. We have removed the statement, here is the revised and clarified system description:

“The system considered here is a nanopatterned square ASI array of stadium-shaped 3D nanoislands comprising four distinct layers. From substrate (SiO₂) to top (Fig 1a): NiFe(30 nm)/Al(35 nm)/NiFe(20 nm)/Al(5 nm). The state of each magnetic layer is independently programmable, with ‘hard’ (30 nm NiFe, lower layer) and ‘soft’ (20 nm NiFe, upper layer) layers switching at higher and lower relative H fields respectively.

Islands are lithographically defined via electron beam lithography and thermal evaporation liftoff process with dimensions of 550 x 140 x 90 nm, with vertex spacing of 125 nm, measured island-end to vertex-centre. A 50 nm lateral displacement in the y direction (Fig1b) between hard and soft NiFe layers induced via shadow deposition for vortex chirality control.

An attractive feature of this design is with a single lithography step, an arbitrary number of layers can be deposited without breaking vacuum.“

Reviewer #3 (Remarks to the Author):

This is an experimental investigation, supported by micromagnetic simulations, into spin waves in an artificial spin ice structure. The structure is composed of multilayered nanoelements, which consist of two permalloy discs of elliptical shape and varying thicknesses. Moreover, the top permalloy layer is shifted by 50 nm in relation to the central position, and the bottom layer. With this expansion, a significant increase in the number of possible microstates is achieved compared to the standard artificial spin ice system. Furthermore, the potential to control various states using an external magnetic field is shown. This system demonstrates a rich spin-wave spectrum that is strongly dependent on the magnetization configuration, including magneto-toroidal states. The authors assert that ultra-strong interlayer magnon-magnon coupling is evident as a result of dipolar interactions. This study is both timely and intriguing, expanding the research on artificial spin ice systems and the application of magnonics. However, prior to the decision on its suitability for publication, improvement is necessary.

We thank the reviewer for their nice assessment of our work, and for the time taken to put together their review. We have substantially improved our manuscript by responding to the thoughtful and constructive points, and the manuscript is now greatly improved as a result.

1. The magnon-magnon coupling is described by the static dipolar interactions between the elements, particularly the Py discs. However, it remains unclear whether and to which extent dynamic dipolar couplings are also involved. It should be noted that photonics, which authors refer to when discussing ultra-strong coupling, involves dynamic coupling between two modes without static effects. In this manuscript, the splitting of acoustic and optic modes seems to be attributed to the mutual modification of the static internal magnetic field of the discs. How reasonable is it to compare it to photonics and name it ultra-strong coupling?

The referee raises a good question about whether the mode hybridisation and generation of acoustic and optical modes arises from static or dynamic dipolar coupling. It is a good question as this system has both static and dynamic dipolar coupling, both of which impact the system in different ways. We have revised our manuscript throughout to make it explicitly clear which component (static or dynamic) of the dipolar field is responsible for each of the results we show. We thank the referee for this opportunity to improve the clarity of our work.

We can break down the impact of the two static and dipolar field components:

Static component of the dipolar coupling:

A static dipolar field is effectively equivalent to the effects of a uniformly applied external magnetic H field, other than it is localised to a specific spatial region of the system.

The effect of a static field is to apply a DC shift to the frequency of a magnon mode. A blueshift if the static field is aligned along the internal magnetisation of the nanomagnet, or a redshift if the static field is aligned anti-parallel to the internal magnetisation.

An example of this effect caused by a uniform external field may be seen in figure 2b) of the revised figure 2:

In the -20 to 0 mT region, the system is in a parallel macrospin state with both layers magnetised in the negative x direction. Here decreasing the magnitude of the -x field redshifts the mode.

In the 22-65 mT region, the system is in a parallel macrospin state with both layers magnetised in the positive x direction. Here increasing the magnitude of the +x field blueshifts the mode.

We can examine the effect of static dipolar field by looking at similar mode redshift/blueshift behaviour when the externally applied field is zero. In the parallel macrospin case, the dipolar field emanating from one layer is aligned opposite to the internal magnetisation of the adjacent layer - hence we expect a mode redshift in this case. In the antiparallel case, the opposite is true and the dipolar field emanating from one layer is aligned parallel to the internal magnetisation of the adjacent layer. This is illustrated schematically in the supplementary information figure copied below:

Figure 10. Schematic of stray field in parallel and antiparallel macrospin states. In the antiparallel (AP) case, dipolar field emanating from one layer sums with the magnetisation in the other layer, increasing the internal field and hence raising the magnon mode frequency. However, outside of the nanoisland the stray fields of each layer cancel each other - hence in the AP nanoisland case the internal dipolar field is strong, and the external dipolar field is weak. In the parallel (P) case, the dipolar field emanating from one layer opposes the magnetisation in the other layer. However, outside of the nanoisland the stray fields of each layer sum - in the P case the external dipolar field is strong.

If we look at our experimental FMR data taken in zero-field, we see that the expected mode red/blueshifts indeed occur as expected, with the antiparallel state (orange trace in figure

below) having a mode frequency of 11.6 GHz, 1 GHz higher than the parallel state (black trace in the figure below) mode frequency of 10.6 GHz. This is shown in figure 2 d) and e):

We have carried out further simulation work to support this. Below we show the simulated (mumax3) spatial profile and magnitude of the static component of the stray dipolar field projected by a single 30 nm NiFe nanoisland (left four columns) and single 20 nm NiFe nanoisland (right four columns). No GHz excitation is applied, no magnon modes are active (simulations are performed at zero effective temperature) so all the stray dipolar field here is static in nature. Each row is a 10 nm z-slice, with the z position of where the adjacent magnetic layer would be highlighted by the upper curly brackets in each case.

The quadrature combined x,y,z stray dipolar field shows peak values at the adjacent magnetic layer position of 60-80 mT (30 nm island) and 35-45 mT (20 nm island):

From Fig 3a) (shown below) we can look at the amount of spatially-uniform externally applied magnetic field required to generate a shift of 1 GHz (the difference in mode resonance between the parallel and antiparallel macrospin states). From a resonance of 11 GHz at 20 mT and a resonance of 12 GHz at 53 mT, we can take the mode gradient in this region as 1 GHz / 33 mT in the case of a uniformly applied external magnetic field.

This is somewhat lower than the dipolar field projected by the nanoislands, but it is the same order of magnitude and as the projected dipolar field is very locally concentrated at the nanoisland ends vs. evenly distributed, it is unsurprising that a higher peak field value is required in the spatially-nonuniform stray dipolar field case to generate the same mode frequency shift as in the spatially-uniform externally applied field case.

Dynamic component of the dipolar coupling:

The case of our ultrastrong magnon-magnon coupling including mode splitting, mode hybridisation and generation of acoustic and optical modes is quite different from the DC mode frequency shifts discussed above. Below we show mumax simulations of the antiparallel state for a range of inter-layer spacing thicknesses, from 250 nm to 20 nm:

At a spacing thickness of 250 nm, there is insufficient dynamic dipolar field coupling to drive hybridisation between magnon modes in the upper and lower magnetic layers. As such, there is no anticrossing, no appreciable magnon-magnon coupling and the positive gradient

mode corresponds to the upper magnetic layer with positive magnetisation (highlighted in purple) and the negative gradient mode corresponds to the lower magnetic layer with negative magnetisation (highlighted in green).

At spacing thicknesses of 80 nm and below, we see very different behaviour. Here, an anticrossing gap (labelled $\Delta\omega$) is generated and the modes hybridise, with each mode branch consisting of a coupled oscillation between upper and lower magnetic layers. The difference in the two mode branches is the phase relationship between the upper and lower magnetic layers. The upper branch (red) is the acoustic mode where both magnetic layers oscillate in-phase, the lower branch (blue) is the optical mode where both magnetic layers oscillate out-of-phase. A suite of evidence showing the optical/acoustic mode character is described in detail in response to reviewers 1 and 2 above.

In order to generate an anti-crossing gap opening and hybridised in-phase/out-of-phase modes, the coupling mechanism responsible must be dynamic. A static coupling mechanism is unable to generate different mode energies or frequencies for different phase relations between the coupled modes as it has no way of providing different responses at the timescale of the mode oscillation, e.g. a temporally static dipolar field has no means of providing mode frequency shifts for modes which are described by the relative phase of their GHz oscillations.

A further piece of evidence here that the dynamic component is responsible for the magnon-magnon coupling is the opening of the anti-crossing frequency gap - a static coupling mechanism can only provide DC frequency shifts and has no way to force the non monotonic curvature of a given mode and hence open a frequency gap.

If the coupling was purely static, the only observed phenomena would be DC redshifts and blueshifts of magnon modes, which would still be permitted to cross and overlap in frequency, similar to the crossing seen in the 250 nm case above. The fact that we see anti-crossing gap opening, acoustic and optical phase relationships and mode hybridisation between the upper and lower magnetic layers is strong evidence that the mechanism driving the ultrastrong magnon-magnon coupling is dynamic dipolar field coupling, eg. coupling via oscillatory GHz components of the dipolar field.

We have added the following statement to the introduction to make the distinction between the static and dynamic components of the dipolar field coupling explicit:

“The strong dipolar coupling involves both static and dynamic components of the dipolar field. Static components are leveraged for GHz-scale mode frequency shifts between microstates and chiral microstate control, while dynamic components enable ultrastrong magnon-magnon coupling between magnons in the upper and lower magnetic layers, mode-hybridisation and generation of anticrossing gaps.”

2. In Fig. 1a, the structure shall be shown with the relative shift of the top layers to be consistent with the structure under investigation.

This is a good suggestion, and we have implemented it. We thank the reviewer for the suggestion.

3. Why the thicker Py layer is called “hard” and the thinner “soft”? Seems to be that the thinner has a larger shape anisotropy.

While the reviewer is correct that the shape anisotropy is larger for the thin layer, the hard/soft designation here refer to the magnitude of the coercive switching field. The coercive field is also related to the thickness of the magnetic nanoisland layer. While fabricating narrower nanoislands in the x,y plane does indeed increase the switching field via the shape anisotropy, for 20 nm thick vs 30 nm thick nanoislands with identical in-plane X,Y dimensions, the 30 nm thick layer has a higher coercive field.

This can be seen in figure 2. The MOKE data shows a large step at the higher switching field, corresponding to the larger volume of the 30 nm layer. Our mumax simulations show that when sweeping field negative to positive and switching from the negative saturated parallel macrospin state to the antiparallel macrospin state, the 30 nm ‘hard’ layer remains unswitched and the 20 nm ‘easy’ layer reverses first:

4. There is the statement “The fabrication parameters are tuned to give the strongest inter-layer dipolar coupling possible while keeping all states stable in zero applied magnetic field ($H_{ext} = 0$).” What does it mean strongest coupling and how it was determined?

As the coupling between magnetic layers here is via dipolar field which drops off with distance as $1/r^3$, the coupling strength is determined by the thickness of the non-magnetic spacer layer, eg. the inter-layer spacing distance between the magnetic layers. The coupling strength may be parameterised via the frequency width of the gap between optical and acoustic modes or by the normalised coupling rate (frequency width of the gap divided by the frequency of the upper mode frequency), which we show using micromagnetic simulation strongly increases as the inter-layer spacing distance is reduced:

As you reduce the spacer layer thickness to small thicknesses (sub 30 nm), the dipolar field of the thicker 30 nm layer becomes strong enough that it overcomes the coercive switching field of the thinner 20 nm layer, and forces the thinner layer to switch into a macrospin state oriented anti-parallel to the magnetisation of the 30 nm layer (Eg the anti-parallel macrospin state). Hence, the only stable state at remanence becomes the anti-parallel macrospin state, and all the rich reconfigurability of this system is lost. Hence, the optimal conditions are such that the coupling strength is maximised (i.e. the inter-layer spacing is as small as possible) while ensuring that the dipolar field of the 30 nm layer does not lock the 20 nm layer into the antiparallel macrospin state at remanence (eg. keeping all 16 states stable in zero applied magnetic field).

To make this clearer in the text, we have amended the statement to:

“The fabrication parameters are tuned to give the strongest inter-layer dipolar coupling possible while keeping all states stable in zero applied magnetic field $H_{Ext} = 0$. Dipolar coupling strength is parameterised by the normalised magnon-magnon coupling rate described below in the discussion of figure 3.”

5. From the caption of Fig. 2f it is not clear meaning of the color map and dots.

These are the negative parallel, antiparallel and positive parallel states. The caption has been amended to include:

“Marker colour-code refers to the same parallel/antiparallel states as in (b).”

6. From fig. 6 a and b it seems that the agreement between measurements and simulation results is far from being perfect. What can be a reason?

We apologise for the potentially confusing subheading labels in this supplementary figure. Figure 6 a and b are both simulations, the ‘experimental’ subheading on 6b was an indication that the Y geometry was the RF field geometry we used in experiment when observing the optical mode. We have revised the subheadings, shown below:

The referee is of course correct that there are differences between experiment and micromagnetic simulation. This is due to a number of reasons, probably the biggest is that micromagnetic simulations show more modes than we can resolve experimentally. This is due in part to the fact that there is zero experimental noise in simulation, so weak/faint modes can be observed without falling below the experimental noise floor. Also, nanoisland shapes are perfect and symmetrical, many of the weaker higher-order modes are highly-sensitive to small imperfections and asymmetries in patterning and these modes become much weaker in real imperfect nanostructures. Also, in experiment we are averaging over roughly 10^8 nanoislands, each with slightly different imperfections, so many fine modes and features are effectively washed out from this averaging process.

In terms of frequency differences between the dominant modes and differences in the coercive switching fields, mumax3 is not taking any thermal effects into account or effects such as edge roughness which can strongly affect nucleation of domain walls and switching dynamics. Also the mode frequencies are strongly dependent on M_s (Saturation magnetisation). We have used a standard value for nanostructured permalloy, it is possible to iterate this number to achieve closer correspondence between frequencies but without a

real experimental measurement to get exact values, we prefer to state that we use a commonly employed value rather than put in somewhat arbitrary material parameter numbers without experimental evidence. There is a similar effect on the exact shape and size of the nanostructures, exactly how elliptical/circular the nanoisland ends are etc. Again, we prefer to keep the simulations as simple as possible, using semi-circular end caps and a rectangular 'main body' of the nanoisland. This gives some frequency mismatch between experimental and simulated modes, but it keeps the study clearer and simpler to reproduce.

In BLS spectra, fig 2f, the high-frequency band as a function of H in the antiparallel state seems to be nonmonotonous, while in the parallel state two intensive bands are visible, which does not seem to be present in simulation results, why?

The BLS results are obtained from much fewer nano-elements than the FMR (a few nanobars in BLS vs. $\sim 10^8$ nanoislands in the FMR experiments).

Arrays of nanofabricated elements always contain some distribution of island sizes, material quality (different saturation magnetisation etc in the case of magnetic nanoelements) and hence a distribution of magnon dynamics. This distribution of nanofabrication imperfections is often termed quenched disorder.

When fabricating very large arrays, such as the 10^8 island arrays in this manuscript, a greater than normal degree of quenched disorder. More subtle behaviours such as exact mode curvatures may be averaged over to a degree.

If we look at the micromagnetic simulations of a an array of perfectly identical bars (see fig 3j below) we actually observe the non-monotonous behaviour seen in the micro-focused BLS data, which is experimentally measured from a diffraction-limited laser spot, eg. sensitive to a single bar or a few bars at most:

The large difference in the number of nanoislands being measured and subsequent effective reduction in quenched disorder over the measured islands is likely the explanation of the non-monotonous behavior of the high-frequency band in the BLS data.

In terms of the two intensive bands, the referee raises a good point that the BLS data has some different features relative to the experimental FMR data. There are a few reasons for this. The primary reason is that the RF field in the BLS experiment is oriented in the

z-direction (out of the plane). In a conventional synthetic antiferromagnet style multi-layer system, no coupling to the optical mode would be possible in this geometry, but as we break the z-direction symmetry via our nanopatterned geometry (we use different NiFe layer thicknesses and the dynamic dipolar field has a non-zero curvature in z, whereas the exchange interaction in a synthetic antiferromagnet thin-film is symmetric in z) we are able to observe the optical mode with a z-direction RF field. This demonstrates some of the additional flexibility of our system relative to thin-film multilayer systems, hence the choice of a z-direction RF field. Below we show a micromagnetic simulation for the case of a z-direction RF field:

Another consequence of choosing a z-direction RF field is that the parallel state (above 48 mT in the above plot) has a higher-order mode at a frequency above that of the main Kittel mode. This mode matches the frequency of the acoustic mode in the antiparallel state (0-48 mT in the above plot), such that the acoustic mode appears to continue into the parallel state when in fact it is a distinct mode but with similar frequency/gradient behaviour. BLS may also exhibit higher sensitivity to some higher-order modes than FMR measurements, hence this higher-order mode is resolved at relatively strong intensity in the BLS data - showing as the two intensive bands in the parallel state.

The FMR experimental data uses an x-direction RF field. From a micromagnetic simulation of this field geometry shown below, one can see that the higher-order modes are of lower intensity relative to the main mode vs. a z-direction RF field-excitation, and as mentioned FMR may be less sensitive to higher-order modes. The combination of these factors mean that the BLS data shows the higher order mode more strongly. The sample used for the BLS experiment is also of slightly different dimensions, and as such relative frequencies vary slightly relative to the FMR data as described in the text.

Below, we show experimental FMR data measured up to a higher maximum frequency (15 GHz). The higher order mode is also visible in the FMR, it is just weaker due to the different RF field direction and difference in measurement method:

The following statement has been added to the main manuscript:

“The BLS data shows a minimum anticrossing gap at a small finite positive field of 7 mT. This matches the behaviour seen in our micromagnetic simulations and occurs as the two magnetic layers have differing thicknesses (equal thickness layers give a zero-field minimum gap). This is clearer in the BLS data as it measures just a few islands, while the FMR data measures a 10^8 nanoisland array where nanopatterning imperfections (often termed ‘quenched disorder’) can average out such effects. The mode seen above the main Kittel mode in the BLS data is a consequence of the BLS being measured using a z-direction RF field, with corresponding z-direction micromagnetic simulations showing similar behaviour (see supplementary figure 6)”.

7. Page 7, probably in the last two paragraphs references to figures shall be corrected. Also on p. 10, second paragraph.

Many thanks for spotting this. We have amended and carried out another round of copy editing for similar issues.

8. The modes identified on p. 8: “An extremely rich spectra is observed, given by a combination of magnetic vortex states (χ -shaped modes, 2-6 GHz). edge-curvature states (dominant edge-curvature mode overlaps with the 2-3 GHz vortex modes, and 3 higher-frequency edge-curvature modes 3-5 GHz)” are very difficult to find in the figure.

We appreciate that the spectra here is complex and quite dense.

To make reading the spectra easier, we have added the below annotated plot to the supplementary information, with a reference to this supplementary figure given when introducing Fig 4i):

Figure 13. Mode assignment for Fig. 4 i) labelling which magnetic textures each mode originates from

We have give reference in the sentence after the spectra are introduced to a prior work where readers can find the vortex modes without the extra edge modes, and from comparison of the two spectra we believe finding the various modes is made easier: “Examples of magnetic vortex magnon spectra without edge-curvature modes also present may be found in a prior work by some authors (Gartside, Jack C., et al. Nature Nanotechnology 17.5 (2022): 460-469).”

9. Which magnetisation components were used to calculate the power spectra?

The combined x,y,z components via the expression $\sqrt{M_x^2+M_y^2+M_z^2}$ are used for the simulated field vs frequency spectra & the spatial power maps.

10. The caption of Fig. 7 in Supl. Mat. Needs to be extended to describe in detail all presented plots.

This figure is being removed.

11. Fig. 11 and 12: The color scale for the demagnetizing field values is missing.

We apologise for the lack of clarity on our part. We have updated the figure caption and added a colourwheel with explanation that we only show the magnetisation direction here, not the magnitude:

Caption: The demagnetisation field between bilayer single islands. (a) shows the mumax3 magnetisation in the antiparallel (AP) and in (b) shows the magnetisation in the parallel (P) configuration. The colourwheel below maps the direction of the demagnetisation field in 3D space with the shading (black to white) showing the z-components and the hue showing the xy-components. The magnitude is not shown in this figure. (c) shows the demagnetisation field in the spacer layer region for the AP configuration. The opposite poles on the two ends generate a strong z demagnetisation field that extends beyond the spacer layer, looping d from the bottom bar on the left and round down into the bar on the right. There is a slight demagnetisation field to the right due to the unequal thickness of the top and bottom bar. (d) shows the demagnetisation field in the P configuration, here, the like poles of the ends of the bar repel and create equal but opposite demagnetisation field in z in the space resulting in a net demagnetisation field in the xy-plane while little in z beyond the bar. The magnitude of the demagnetisation field in both the antiparallel (AP) and parallel (P) in the bottom, gap and top layers is shown in (e, f) respectively.

Caption: The demagnetisation field between bilayer ASI. First row shows the demagnetisation field direction as a heatmap with the shade (black into the plane) showing the z-component and hue showing the xy-component. The second row shows the magnitude of the demagnetisation field and the quiver plot shows the demagnetisation field in the xy-directions. (a) shows the mumax3 simulated demagnetisation field integrated in the spacer between the top and bottom layer (30 – 60 nm) while in the AP state. There is strong coupling between the opposite poles in the bilayer generating a strong z-demagnetisation field and very little lateral coupling as shown by (d). This breaks the symmetry leading to strong coupling between the two left bilayer islands with the same downward demagnetisation field and vice versa for the right side. (b) shows the demagnetisation field in the P state where the like charges make the demagnetisation field much more similar to what is seen in single layer ASI (c). The nanobars are laterally coupled shown in (e) and (f)

The arrows show the net moment of the bars (ie the magnetisation of the bottom layer).

REVIEWER COMMENTS

Reviewer #1 (Remarks to the Author):

I am satisfied with the author's responses to all referees and the changes made to the manuscript. I recommend publication in its present form.

Reviewer #2 (Remarks to the Author):

Dear Editor,

I more carefully read the rebuttal letter now after my first quick scan. In most cases the authors agree with my criticism and then go in a very detailed discussion why the original claims can still be valid if just softened in a certain manner. They did so throughout the entire manuscript, but still left the title and abstract and hence are still overselling and dramatizing the results.

My previous recommendation to reject the paper was mainly based on the fact, that the paper is stitching together three different pieces, which could each be a separate publication. But quantity doesn't make quality, especially if it's for a high impact journal addressing a broad audience. The authors fixed most of the problems I was mentioning but still the paper lacks significance (still overselling a lot) and is way too complicated and long for what I believe a good article in Nature Communications should be.

A point by point reply doesn't make sense for me here. If it was requested, it would take a long time to work through the 35 pages long rebuttal letter and all the new results shown there and thus, severely delay the review process. My recommendation still is "reject" and publish the paper either in two or three publications in more specialized journals or use it in combination with the supplementary and the data shown in the rebuttal to write a more detailed review article on the topic.

Reviewer #3 (Remarks to the Author):

The authors made a major revision of the manuscript, significantly extended its content with new results and discussion, and answered all the questions of the reviewers. In my opinion, it is a well-written paper with interesting physics, which meets the publication criteria of Nature Communication.

I have one more question for the authors to consider. In Figs. 3a-d there is a nice presentation of the increasing coupling between acoustic and optical modes with decreasing separation between the layers. However, there is an additional line visible, with particularly strong intensity in Fig. 3c, where it is located between the acoustic and optical modes. Its frequency seems to depend on the separation as well. What kind of excitation is associated with this line and is it also coupled to acoustic and/or optical mode?

RESPONSE LETTER Ultrastrong Magnon-Magnon Coupling and Chiral Spin-Texture Control in a Dipolar 3D Multilayered Artificial Spin-Vortex Ice

Reviewer #1 (Remarks to the Author):

I am satisfied with the author's responses to all referees and the changes made to the manuscript. I recommend publication in its present form.

We thank the reviewer for their careful consideration of our work and the recommendation to publish.

Reviewer #2 (Remarks to the Author):

Dear Editor,

I more carefully read the rebuttal letter now after my first quick scan. In most cases the authors agree with my criticism and then go in a very detailed discussion why the original claims can still be valid if just softened in a certain manner. They did so throughout the entire manuscript, but still left the title and abstract and hence are still overselling and dramatizing the results.

We thank the reviewer for taking the time to read our revised manuscript.

We appreciate the reviewer's concerns, but believe we have demonstrated ultrastrong magnon-magnon coupling both experimentally and in simulations. We have shown the acoustic/optical character of our hybridised modes via simulated magnetisation vs. time traces and spatial phase maps, and experimentally via the optical mode only appearing when applying RF parallel to the DC field in a parametric pumping geometry. The gradual opening of the anticrossing gap when reducing spacer thickness in simulations further supports our claims.

My previous recommendation to reject the paper was mainly based on the fact, that the paper is stitching together three different pieces, which could each be a separate publication. But quantity doesn't make quality, especially if it's for a high impact journal addressing a broad audience. The authors fixed most of the problems I was mentioning but still the paper lacks significance (still overselling a lot) and is way too complicated and long for what I believe a good article in Nature Communications should be.

We thank the reviewer for their comments. The aim of this paper is to introduce a new architecture of three-dimensional artificial spin ice nanomagnetic array that enables a range of behaviours which are not observed in its two-dimensional counterparts, while maintaining the relative ease and accessibility of conventional two-dimensional arrays. As such, including these results together makes good narrative sense to us.

A point by point reply doesn't make sense for me here. If it was requested, it would take a long time to work through the 35 pages long rebuttal letter and all the new results shown there and thus, severely delay the review process. My recommendation still is "reject" and publish the paper either in two or three publications in more specialized journals or use it in combination with the supplementary and the data shown in the rebuttal to write a more detailed review article on the topic.

We appreciate that our response letter was long and thank the reviewer for reading it. Our intention was to thoroughly address all points raised, and we endeavoured to perform a range of additional studies to do this. Some of the same points were raised from several reviewers and were repeated in the letter to allow ease of reading for each reviewer, adding to the length. Much of the response letter included new figures which we believe greatly strengthen the utility and clarity of the manuscript.

Reviewer #3 (Remarks to the Author):

The authors made a major revision of the manuscript, significantly extended its content with new results and discussion, and answered all the questions of the reviewers. In my opinion, it is a well-written paper with interesting physics, which meets the publication criteria of Nature Communication.

I have one more question for the authors to consider. In Figs. 3a-d there is a nice presentation of the increasing coupling between acoustic and optical modes with decreasing separation between the layers. However, there is an additional line visible, with particularly strong intensity in Fig. 3c, where it is located between the acoustic and optical modes. Its frequency seems to depend on the separation as well. What kind of excitation is associated with this line and is it also coupled to acoustic and/or optical mode?

We thank the reviewer for raising this point, we do indeed see these modes in simulation and experiment.

We have performed further analysis via micromagnetic simulation of the 40 nm spacer-layer case, with the results shown in a new supplementary figure below. Our analysis is that these are higher order modes of the optical and acoustic modes as evidenced by the mode gradients, labelled as acoustic' and optical'. We perform a sinusoidal RF excitation field at the acoustic mode frequency and observe a beating frequency of 217 MHz, with the same beating envelope in time observed in both modes. The beating frequency is close to the gap frequency between the acoustic and optical' modes, and alongside the matching beating envelope this is suggestive of coupling between the acoustic and optical' modes which is an interesting prospect. Confirmation of this coupling would require some additional studies, and at this stage this is beyond the scope of the current work.

We have included the new figure as supplementary information and include the following in the body of the main text:

“Below 40 nm spacer thickness extra higher order modes appear in the spectra whose spatial power and phase maps are shown in supplementary figure 16.”